# Induction of synapse formation by *de novo* neurotransmitter synthesis

Scott R. Burlingham[1,6], Nicole F. Wong [2,6], Lindsay Peterkin[1,6], Lily Lubow[1], Carolina Dos Santos Passos[1], Orion Benner[1], Michael Ghebrial [3], Thomas P. Cast [1], Matthew A. Xu-Friedman [2✉], Thomas C. Südhof [4✉] & Soham Chanda [1,4,5✉]

A vital question in neuroscience is how neurons align their postsynaptic structures with presynaptic release sites. Although synaptic adhesion proteins are known to contribute in this process, the role of neurotransmitters remains unclear. Here we inquire whether *de novo* biosynthesis and vesicular release of a noncanonical transmitter can facilitate the assembly of its corresponding postsynapses. We demonstrate that, in both stem cell-derived human neurons as well as in vivo mouse neurons of purely glutamatergic identity, ectopic expression of GABA-synthesis enzymes and vesicular transporters is sufficient to both produce GABA from ambient glutamate and transmit it from presynaptic terminals. This enables efficient accumulation and consistent activation of postsynaptic GABA$_A$ receptors, and generates fully functional GABAergic synapses that operate in parallel but independently of their glutamatergic counterparts. These findings suggest that presynaptic release of a neurotransmitter itself can signal the organization of relevant postsynaptic apparatus, which could be directly modified to reprogram the synapse identity of neurons.

[1] Biochemistry & Molecular Biology, Colorado State University, Fort Collins, CO, USA. [2] Biological Sciences, State University of New York at Buffalo, Buffalo, NY, USA. [3] Biological Science, California State University Fullerton, Fullerton, CA, USA. [4] Molecular & Cellular Physiology, Stanford University School of Medicine, Stanford, CA, USA. [5] Molecular, Cellular & Integrated Neurosciences, Colorado State University, Fort Collins, CO, USA. [6] These authors contributed equally: Scott R. Burlingham, Nicole F. Wong, Lindsay Peterkin. ✉email: mx@buffalo.edu; tcs1@stanford.edu; soham.chanda@colostate.edu

Neurons communicate with each other via specialized structures called synapses. How synapses establish and maintain their identity remains largely unclear. According to one theory, synaptic cell adhesion molecules (SAMs) trigger synapse formation by promoting trans-synaptic interactions between pre- and postsynaptic components[1–3]. In support of this hypothesis, ectopic expression of SAMs can respectively enhance or induce synaptogenesis in both neurons and non-neuronal cells[4–7]. Moreover, individual genetic deletions of some SAMs were also reported to decrease synapse numbers by variable degrees, although they did not completely eliminate synapse assembly[8–10].

Interestingly, despite these few instances, constitutive or conditional knock-out (KO) models for the vast majority of SAMs do not exhibit any large-scale impairments in synapse formation and only affect their functional maturation, which occasionally might lead to a subsequent loss of synapses over time[11–15]. Furthermore, SAM-dependent mechanisms are yet to explain the productions of different types of synapses, because several postsynaptic SAMs specifically localized at either glutamatergic or γ-aminobutyric acid (GABA) -ergic principal synapses can often interact with common presynaptic binding partners that are similarly distributed at both synapse types[16–18]. Hence, alternative cellular signals other than SAMs could be either primarily or simultaneously required for synaptogenesis and reliable alignment of complementary synaptic apparatus.

A second model of synaptogenesis implies that neurotransmitter release can directly modulate this process. In response to various transmitters produced in the presynaptic terminals, their postsynaptic compartments recruit distinct classes of receptors that confer functional properties to synapses. This theory is further strengthened by studies showing that deletion of GABA$_A$ receptors (GABA$_A$Rs) can impair both the morphology and target specificity of a subset of GABAergic synapses[19,20]. Additional evidence for transmitter-dependent postsynaptic arrangements was obtained from recent observations that different co-transmitters synthesized within a single neuron can often become segregated at independent presynaptic terminals that contact distinct postsynaptic cell populations[21–23]. Furthermore, some neurons can even switch between transmitter types in an activity-dependent manner, which in turn alters their corresponding receptor levels and compositions at postsynapses[24,25].

Perhaps the most convincing case for transmitter-induced synaptogenesis appears from two seminal studies demonstrating that rapid photolysis of 'caged' glutamate and GABA near dendritic branches can cause local accumulation of postsynaptic receptors and scaffold proteins, resulting in immature synapse formation that can eventually integrate into existing neural circuits[26,27]. This phenomenon was also successfully reproduced in different neural subtypes located at various brain regions of a broad age-range of animals[28–30]. However, it remains unknown (i) whether such mechanisms can operate during physiologically relevant presynaptic neurotransmitter release, (ii) if these transmitter-induced nascent synapses undergo further morphological and/or functional maturation, (iii) whether the transmitter identity of a neuron itself could be deliberately manipulated using exogenous factors, (iv) if changing the neurotransmitter released might directly lead to the production of different synapse types, and (v) whether these transmitter-induced synapses could also develop in vivo, especially in live animals. Addressing these questions might allow one to understand the fundamental principles of how synapses form and acquire their identities.

In this current study, we set out to determine whether ectopic expression of exogenous presynaptic enzymes and vesicular transporters can drive both the biosynthesis as well as synaptic release of an alternative transmitter in lineage-committed neurons, and initiate the formation of functional postsynapses of a different kind. We found that a combinatorial overexpression of three proteins, i.e. vGAT, GAD65, and GAD67, can adequately synthesize and transmit GABA even from exclusively glutamatergic neurons, which trigger an efficient production of morphologically and functionally mature GABAergic output synapses, both in vitro and in vivo.

## Results

**GABAergic factors absent in glutamatergic neurons**. We first aimed to understand how glutamatergic neurons protect their transmitter identity at synaptic outputs and prevent other specifications, e.g. GABAergic programs. To this end, we employed a previously established model system that rapidly generates pure glutamatergic neurons from human embryonic stem (ES, e.g. H1-line) cells by forced expression of a single transcription factor Neurogenin-2 (i.e. Ngn2; Fig. 1a)[31]. Voltage-clamp recordings (holding potential, $V_{hold} = -70$ mV) at post-induction day ≈ 56–60 detected robust spontaneous postsynaptic currents (sPSCs) with recurring network activities (Supplementary Fig. S2a). These synaptic events were predominantly comprised of excitatory sPSCs (i.e. sEPSCs) since they could be readily abolished by acute application of an AMPA-receptor (AMPAR) antagonist Cyanquixaline (CNQX) but not by GABA$_A$R antagonist Picrotoxin (PTX), thus implying that Ngn2-neurons lack either postsynaptic GABA$_A$Rs, presynaptic GABA release, or both (Supplementary Fig. S1a). However, puff-perfusions of exogenous agonists under the same experimental conditions revealed the existence of fully functional and PTX-sensitive GABA$_A$Rs in addition to AMPARs, indicating that these neurons only transmit glutamate but likely not GABA from their presynaptic terminals (Supplementary Fig. S1b).

In order to identify pre- or postsynaptic machineries that are essential for GABAergic neurotransmission but possibly missing in Ngn2-only cells, we next inspected RNA-sequencing results previously obtained by us[32]. We noticed the presence of various GABA$_A$R subunits, as well as major inhibitory postsynaptic scaffolding molecules (e.g. Gephyrin and Collybistin), but only minimal expression of the Glutamate Decarboxylases (i.e. GAD65 and GAD67) or vesicular GABA Transporter (vGAT) (Supplementary Fig. S1c). These cells, however, had substantial expression of enzymes associated with both glutamate biosynthesis and vesicular loading (i.e. Glutaminase and vGLUT1/2), again confirming their glutamatergic identity (Supplementary Fig. S1c). Thus, although postsynaptic cell population might contain necessary components for functional GABA$_A$R assembly, Ngn2-neurons are largely deficient of presynaptic enzymes for GABA synthesis and vesicular transmission.

**Induction of GABAergic currents by presynaptic enzymes**. We next asked if forced expression of GAD65, GAD67, and/or vGAT in glutamatergic Ngn2-cells can synthesize GABA from ambient glutamate, and package it into synaptic vesicles for subsequent release. We cloned human cDNAs encoding GAD65, GAD67, and vGAT under human Synapsin-1 promoter, made lentivirus particles from the constructs, and co-infected Ngn2-neurons with them (Fig. 1a). At post-induction days 56–60, immunostainings for dendritic MAP2 and synaptic marker Synapsin revealed substantial synapse formation (Fig. 1b). Interestingly, electrophysiological recordings from neurons co-expressing either GAD65 + vGAT or GAD67 + vGAT, but neither factors alone, significantly decreased the frequency of sPSCs with fast τ-decay without affecting their amplitude, and simultaneously increased both the frequency and amplitude of sPSCs with slower τ-decay that were mostly absent in Ngn2-only neurons (Fig. 1c–f).

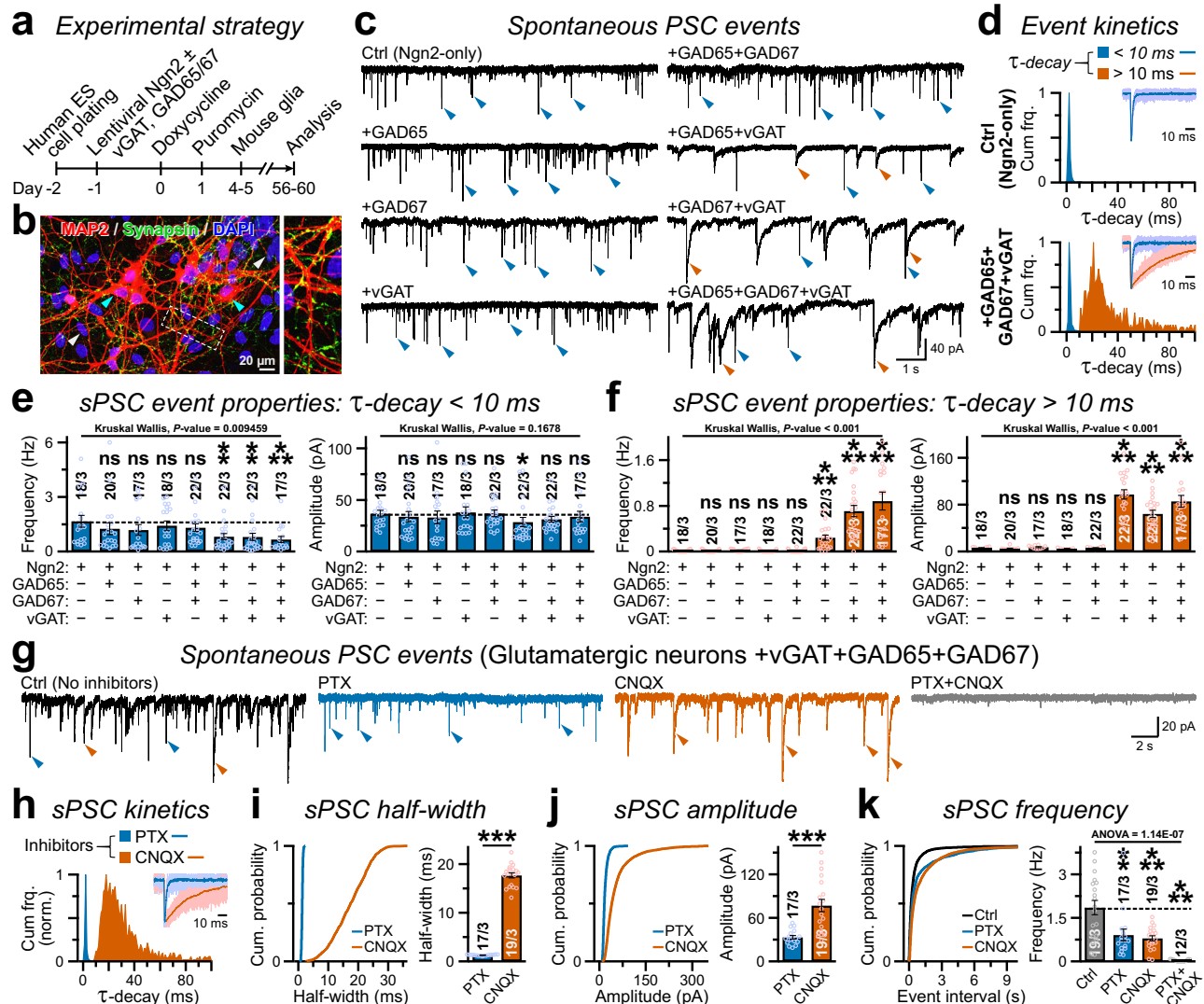

**Fig. 1 Enzymatic synthesis and synaptic release of GABA by defined factors. a** Neurogenesis was induced in H1-ES cells by lentiviral Ngn2 expression, co-infected with additional viruses encoding vGAT, GAD65, and/or GAD67; neurons were co-cultured with mouse glia and analyzed at day 56–60. **b** An example image of Ngn2-induced human neurons (cyan arrowheads) co-transduced with V57 factors, immunolabeled for MAP2, Synapsin, and stained for nuclear DAPI (white arrowheads). The MAP2-negative and DAPI-stained population indicate co-cultured mouse glial cells. Inset, dotted box magnified on right. **c, d** Sample traces of sPSCs recorded from indicated conditions **c**, and normalized cumulative frequency of τ-decays **d** for fast vs. slow (blue vs. red arrowheads) events. Insets in **d**, example waveforms of 10 scaled and overlaid sPSCs (light shade) with corresponding averages (dark shade) for control (top) vs. V57 (bottom). **e, f** Average frequency (left) and amplitude (right) of sPSC events with fast (**e**, τ-decay < 10 ms) vs. slow (**f**, τ-decay > 10 ms) decay kinetics, as recorded from human neurons expressing indicated factor combinations. **g, h** Representative traces **g** and cumulative histogram of τ-decay **h** of sPSCs recorded from Ngn2-neurons co-expressing V57 factors, before (Ctrl) and after acute treatments with PTX and/or CNQX (as annotated). **i–k**. Cumulative probabilities (left) and average values (right) of sPSC half-width (**i**), amplitude (**j**), and event frequency (**k**), measured in the absence (Ctrl) or presence of inhibitors, PTX, CNQX, or both PTX + CNQX. All data are presented as means ± SEM, with number of cells patched / independent batches. Individual data-points are provided as color-matched open circles. For panels **e**, **f**, statistical significance was evaluated by Kruskal-Wallis test paired with post-hoc nonparametric Mann-Whitney U-test using Bonferroni correction (see Source Data). For **i**, **j**, **k**, Skewness and Kurtosis values (-2 >≈ and ≈ < 2) suggested normal distribution, as statistical significance was weighed by two-tailed, unpaired, Student's t-test, with \*\*\*P < 0.005; \*\*P < 0.01; \*P < 0.05; ns = not significant, P > 0.05. Multiple groups in panel **k** were also compared by an analysis of variance (one-way ANOVA) paired with post-hoc Tukey-Kramer test, and corresponding P-values were reported.

Therefore, co-expression of GABA-synthesis enzymes along with GABA vesicular transporter produced sPSC events likely of a different identity than typical EPSCs. The greatest effects observed were for vGAT + GAD65 + GAD67 combination (termed V57 hereafter, Supplementary Fig. S2b).

To determine the identities of sPSC events with fast vs. slow τ-decays, we next applied receptor blockers in Ngn2-cells co-expressing V57 factors. Acute treatments of CNQX vs. PTX

selectively and respectively prevented the fast vs. slow sPSCs with smaller vs. larger half-widths, suggesting that they correspondingly represent excitatory vs. inhibitory synaptic currents (i.e., EPSCs vs. IPSCs; Fig. 1g–i). The spontaneous IPSCs exhibited considerably different average amplitudes, could be inhibited independently from the EPSCs, and only the application of both PTX + CNQX but neither drug alone was able to eliminate all sPSC events (Fig. 1j, k). These results indicate that V57 factors

can successfully prompt the formation of functional GABAergic outputs in glutamatergic human neurons.

**Action potential-dependent and -independent GABA release**. Since sPSCs could be a mixture of both action potential (AP)-independent miniature postsynaptic currents (mPSCs) and AP-dependent network activities, we sought to further characterize their relative contributions in V57-induced GABA release. In current-clamp mode, the Ngn2-neurons co-expressing V57 (termed NV57) displayed repetitive baseline APs, that were readily abolished by tetrodotoxin (TTX) revealing subthreshold depolarizations indicative of AP-independent authentic miniature postsynaptic potentials (i.e., mPSPs; Fig. 2a, b). Accordingly, in voltage-clamp recordings, acute TTX application effectively reduced the amplitude and frequency of both fast and slow sPSCs, but did not completely eliminate all events (Fig. 2c–e). A successive CNQX treatment abolished AMPAR-mediated fast excitatory mEPSCs, and illustrated the co-existence of TTX-insensitive GABA$_A$R-driven inhibitory mIPSCs with slower τ-decays and wider half-widths, that could be consequently silenced by PTX treatment (Fig. 2f–h). Thus, V57-induced presynaptic terminals exhibited spontaneous GABA release in both AP-dependent as well as AP-independent manners, successfully activating the postsynaptic GABA$_A$Rs.

To examine whether GABA can be released by large-scale presynaptic activities, we stimulated the neurons with a field-electrode and measured evoked PSCs. Once again, acute CNQX application detected the presence of prominent evoked IPSC components, with a coefficient-of-variation (CV) equivalent to that of evoked EPSCs, which could be subsequently abolished by PTX treatment (Fig. 2i). In addition, when activated by a train of high-frequency stimulations or repetitive pairs-of-pulses, these evoked IPSCs featured classic short-term plasticity with strong synaptic depression, as was also observed for the evoked EPSCs (Fig. 2j). Therefore, V57-induced noncanonical GABAergic presynapses gained release properties that were comparable to glutamatergic presynapses normally produced by the Ngn2-neurons, presumably due to sharing of common release machineries.

**Effects on synapse morphology**. We inquired if V57 factors enabled the formation of new synaptic structures in Ngn2-cells or remodeled potential glutamatergic synapses into a GABAergic fate. To test that, we immunostained the cells with specific presynaptic antibodies (Supplementary Fig. S3a, b). We did not detect any defects in cell density or overall neurite outgrowth (Supplementary Fig. S4a–c). V57 transduction did not alter the count or morphology of total synapses labeled by Synapsin antibody, and caused only a minor increment in the vGLUT-positive excitatory presynapse size without changing their density (Fig. 3a, b). However, overexpressed vGAT and GAD65/67 factors primarily organized in a clustered pattern, substantially enhancing the size and numbers of GABAergic presynapses as evidenced by their elaborate distribution along dendrites, which was virtually absent in Ngn2-only neurons (Fig. 3c, d).

Especially in V57 cells, co-labeling with vGLUT and vGAT antibodies demonstrated an intricate co-existence of both puncta, which occasionally appeared to overlap in thick z-projected images (Fig. 3e). To explore if synapses of different identities can form within the same presynaptic regions, we utilized super-resolution microscopy. We found that most co-positive puncta included vGLUT vs. vGAT signals that mainly originated from different focal planes, and could be further resolved in x/z or y/z dimensions (Supplementary Fig. S5a). Further analysis of thinner single optical sections suggested that the majority of synaptic puncta

in V57 condition comprised either vGLUT or vGAT signals, and not both (Fig. 3f). Therefore, glutamatergic vs. GABAergic presynapses mostly produced spatially segregated release sites.

To further evaluate any effects on postsynaptic organization, we monitored the distributions of glutamatergic and GABAergic postsynapse markers, e.g. Homer and Gephyrin (Supplementary Fig. S3a, b). We noticed that V57 co-transduction respectively decreased vs. increased the numbers of Homer vs. Gephyrin-positive puncta without affecting their sizes, both in the dendritic segments as well as the perisomatic regions of Ngn2-cells (Fig. 3g, h, and Supplementary Fig. S6a). NV57 neurons also displayed substantial appositions between postsynaptic Gephyrin and cell-surface GABA$_A$R clusters with presynaptic vGAT and GAD65/67 puncta (Supplementary Fig. S6b–d). Despite these rearrangements in synaptic specifications with significant elevation of GABAergic features (Fig. 3i), the V57 condition continued to show limited co-localization between Homer and Gephyrin clusters, that featured distinct shapes and occupied different physical positions (Fig. 3j, and Supplementary Fig. S5b). These results implied that V57-induced GABA release from presynaptic terminals can likely enhance the postsynaptic accumulation of Gephyrin and GABA$_A$R, that were already expressed in Ngn2-only neurons. Nevertheless, these GABAergic synapses assemble separately, i.e. physically isolated, from their glutamatergic counterparts.

**Developmental time-course of synapse maturation**. To assess how early these induced synapses acquire morphological identity and develop functional properties, we analyzed NV57-neurons at different developmental time-points (Fig. 4a). During post-induction day 15–60, the cells matured gradually with a steady rise in membrane capacitance ($C_m$), reduction in input resistance ($R_m$), and increase in overall synapse formation as visualized by Synapsin immunolabeling (Fig. 4b, c). Next, to monitor the relative maturation kinetics of glutamatergic vs. GABAergic synapses, we made patch-clamp recordings. We noticed that the frequencies and amplitudes of both AMPAR-mediated sEPSCs and GABA$_A$R-mediated sIPSCs increased periodically during this timeframe (Fig. 4d, e). Similar maturation kinetics were also observed for both evoked EPSC and IPSC, in terms of their strength and success rates (Fig. 4f, g). In agreement, immunostaining with vGLUT and vGAT antibodies also revealed that both glutamatergic and GABAergic synapses start forming as early as day 15 and continue to increase at day 30, as their density and size tend to saturate around day 45–60 (Fig. 4h, i). Moreover, at day 60, both vGLUT vs. vGAT clusters as well as Homer vs. Gephyrin clusters individually occupied only a fraction of total Synapsin signals, again supporting the notion that synapses of different identities are mostly segregated (Fig. 4j, k). Taken together, these findings suggest that V57-induced GABAergic synapses mature concurrently with glutamatergic synapses, but stabilize independently from each other.

**Activity dependence of GABAergic synapses**. Since the production of GABAergic synapses progressed in parallel with their functional maturation, we asked if GABA-dependent activation of either GABA$_A$R or GABA$_B$R could promote the formation and/or long-term stability of these induced synapses. To probe that, we carried out chronic treatments with either GABA$_A$R antagonist PTX or GABA$_B$R antagonist CGP55845, half-exchanged the media with drugs every other day starting from day 4 to 5, and analyzed the cells on day 56–60 (Fig. 5a). We found that PTX but not CGP application considerably reduced the numbers of both vGAT and Gephyrin puncta without affecting their sizes (Fig. 5b, c). Chronic PTX-treatment did not impair neuronal survival or their dendritic arborization, but diminished the apposition

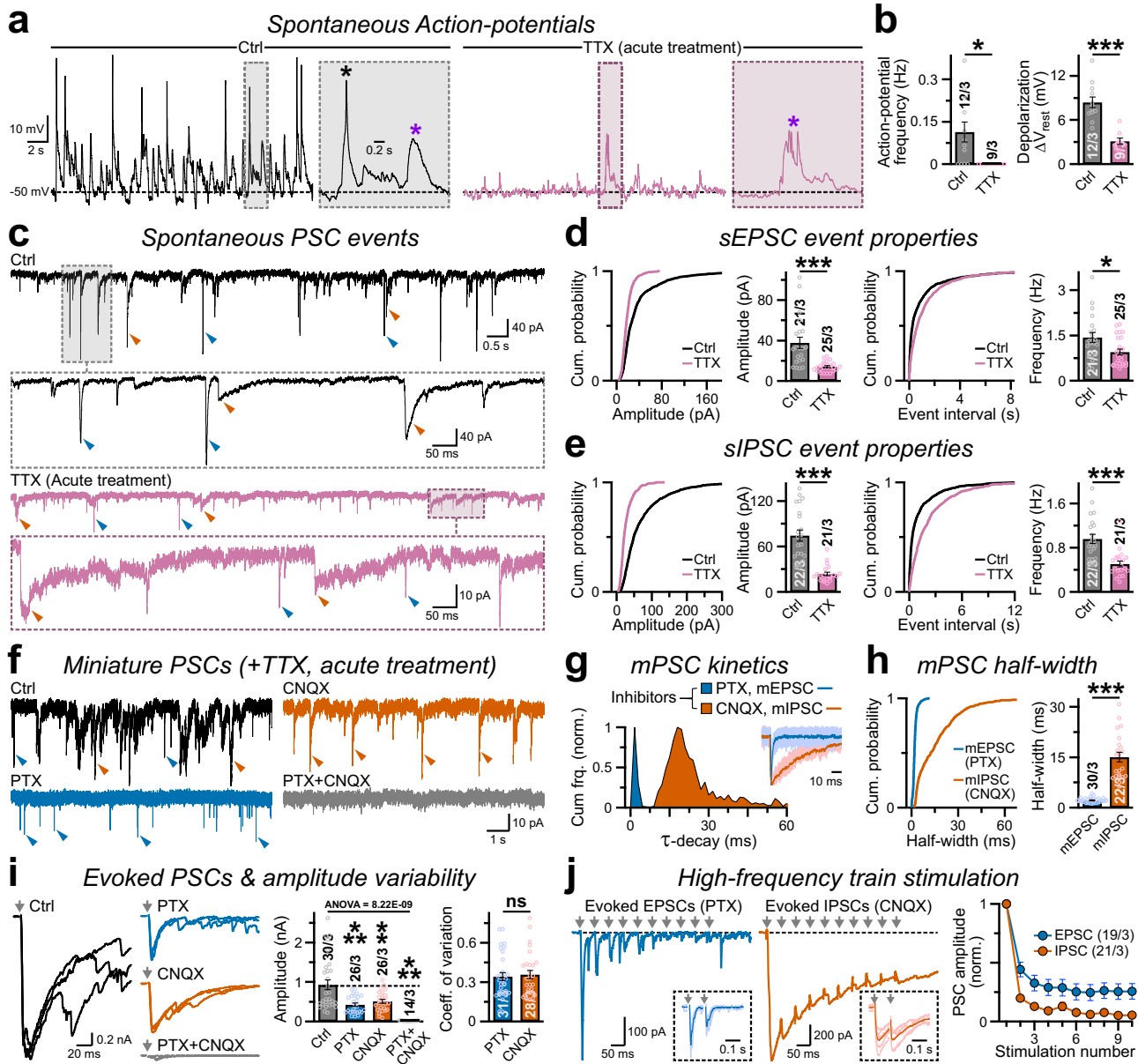

**Fig. 2 V57-induced synapses exhibit both miniature and AP-evoked GABA release. a** Current-clamp recordings of spontaneous APs, in the absence (Ctrl, left) or presence of TTX (right). Insets, magnified areas of the boxed regions with AP (black asterisk) or subthreshold depolarization (purple asterisk). **b** Average frequency of spontaneous AP (left) or total depolarization (right), without (Ctrl) or with TTX. **c**. Representative traces of sPSCs (boxed areas magnified below) recorded in voltage-clamp mode from Ctrl condition vs. acute treatments of TTX, arrowheads pointing at events with fast (blue) vs. slow (red) τ-decay. **d**, **e** Probability plots and averages of event amplitude (left) and frequency (right) for sEPSC **d** or sIPSC **e**. **f** Representative traces of mPSCs recorded in the presence of TTX only (Ctrl), or after co-application of either CNQX, or PTX, or both (CNQX + PTX). Arrowheads indicate fast EPSC (blue) vs. slow IPSC (red) events. **g**, **h** Normalized frequency of τ-decay for miniature PSC events **g**; Inset = 10 superimposed sample traces (light shades) and their corresponding averages overlaid (dark shades). Cumulative probability plots of event half-widths with summary graphs **h**, for PTX-resistance mEPSCs (blue) vs. CNQX-resistant mIPSCs (red). **i** Overlaid representative waveforms (left, stimulus artifacts replaced by arrows), average amplitudes (middle), and CV (right) of PSCs evoked by > = 3 consecutive presynaptic pulses, as recorded from indicated conditions. **j** Example traces (left) of EPSCs and IPSCs evoked by train of pulses (arrows); Insets are successive paired stimulations of presynaptic inputs with Δt = 50 ms, presented as superimposed trials (6 consecutive paired-pulses, light shades) with average traces (dark shades). All PSC amplitudes normalized to corresponding 1st pulse (right). Both EPSCs and IPSCs manifested significant synaptic depression, but of a different magnitude possibly due to different extents of desensitization and/or saturation of postsynaptic AMPARs vs. GABA$_A$Rs, and therefore, could not be directly compared as a proxy for presynaptic release probabilities. All numerical data are means ± SEM, with total number of neurons recorded/experimental batches, and data-points plotted as open circles. Statistical significances were evaluated either by two-tailed, unpaired, Student's t-test (Skewness and Kurtosis values −2 > ≈ and ≈< 2), or two-sided, nonparametric Mann-Whitney U-test, for ***$P < 0.005$; *$P < 0.05$; ns = not significant, $P > 0.05$. Multiple groups were compared by one-way ANOVA (**i**, with $P$-value).

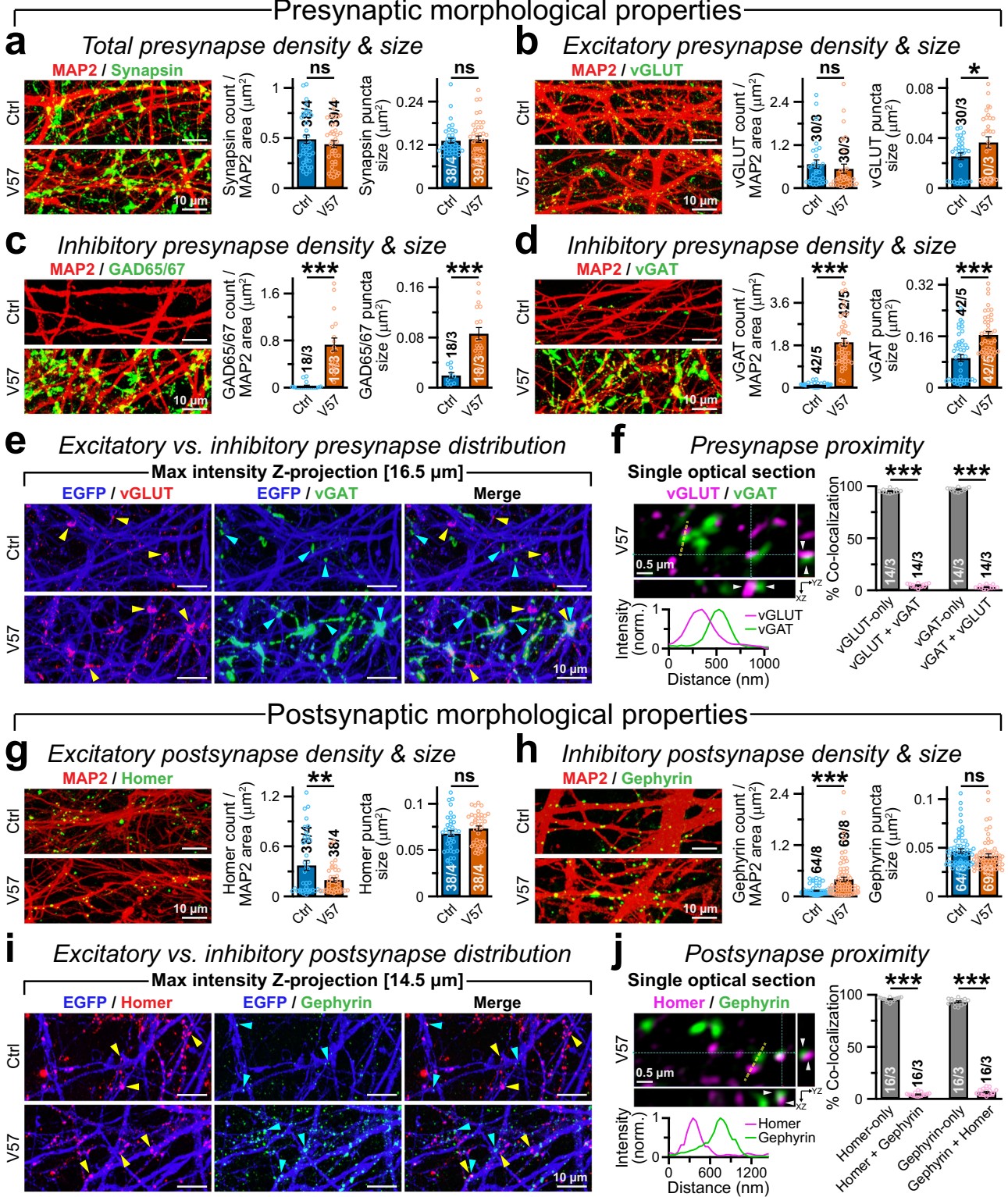

between vGAT and Gephyrin signals (Supplementary Fig. S7a–c), indicating specific defects in GABAergic synapse morphology.

To further evaluate any effects on the functional properties of GABAergic synapses, we examined the surface and synaptic localization of GABA$_A$Rs. We washed out PTX and executed voltage-clamp recordings in drug-free media. Consistent with the reduction in GABAergic synapse numbers, PTX-treated neurons illustrated significant deficit in mIPSC frequency without

changing their amplitude, event kinetics, or intrinsic cell-membrane properties (Fig. 5d, and Supplementary Fig. S7d, e). To assess if this phenotype was caused due to lower surface trafficking of GABA$_A$Rs, we puff-applied exogenous GABA by pressure-perfusion, but failed to notice any substantial effects on GABA$_A$R charge-transfer, implying similar level of receptors at the cell surface (Fig. 5e). However, long-term PTX-treatment decreased the density of dendritic GABA$_A$R clusters

**Fig. 3 V57 factors alter synapse morphology and identity. a** Sample images (left) of Ngn2-only vs. NV57 neurons immunostained with pan-synaptic marker Synapsin. Average values (right) of Synapsin puncta density normalized by dendritic MAP2 area, and puncta size. **b–d** Similar immunostainings as **a**, except for glutamatergic presynapse marker vGLUT **b**, and GABAergic presynapse markers GAD65/GAD67 **c**, or vGAT **d**. **e** Representative images of EGFP-labeled dendritic branches from Ctrl (Ngn2-only, top row) vs. V57 neurons (bottom row) co-labeled for vGLUT and vGAT (yellow and cyan arrowheads, respectively); maximum intensity z-projections highlight an intricate distribution of both synapse types, especially in V57 condition. **f** Super-resolution image (left) of a single optical section from V57 condition depicts minimal co-localization between vGLUT and vGAT signals; grey arrowheads point at partially overlapping signals (cyan crosshair) resolved in x/z or y/z axis; intensity profile shows peak separation between signals from a region-of-interest (yellow dotted line). Mander's coefficients (right) of co-localization between vGLUT and vGAT were plotted. **g**, **h** Same as **a**, but for glutamatergic (Homer, **g**) or GABAergic (Gephyrin, **h**) postsynapse markers. **i** Same as **e**, except for co-existence of elaborate Homer and Gephyrin puncta, especially in V57 condition. **j** Same as **f**, but for postsynaptic Homer and Gephyrin puncta that similarly illustrate minimal co-localization. Average values on bar-graphs represent means ± SEM for number of field-of-views analyzed/independent batches. Individual data-points are provided as color-coded open circles. Statistical significance was weighed by two-tailed, unpaired, Student's t-test (for Skewness and Kurtosis values $-2 > \approx$ and $\approx < 2$), or two-sided, nonparametric Mann-Whitney U-test (Source Data), with ***$P < 0.005$; **$P < 0.01$; *$P < 0.05$; ns = not significant, $P > 0.05$.

---

without affecting their sizes (Fig. 5f). Thus, persistent activation of GABA$_A$Rs but not GABA$_B$Rs regulates the developmental properties and/or maintenance of V57-induced GABAergic synapses.

**Reproducibility of GABA-induced synaptogenesis.** We next inquired whether V57-mediated GABAergic phenotypes could be reproduced when human neurons are differentiated from stem cells other than H1-ES cells. To test this idea, we took advantage of an isogenic induced pluripotent stem (iPS) cell line that stably expressed Ngn2-transgene upon doxycycline induction and generated human neurons with high efficacy, which were essentially glutamatergic and also lacked endogenous vGAT and GAD65/67 expression[33]. Immediately after neural induction, we infected them with V57 viruses, co-cultured with glia, and characterized them at day 35–42 when they attained extensive morphology (Fig. 6a, and Supplementary Fig. S8a). The population showed considerable amounts of vGAT, GAD65, and GAD67 mRNA transduction, and protein localization along their elaborate dendritic arbors (Fig. 6b, c, and Supplementary Fig. S8b–d). These vGAT-enriched synaptic structures also recruited major GABAergic postsynaptic SAM, e.g. Neuroligin-2 (Supplementary Fig. S8e)[12,34,35].

Similar to the H1-ES cell derived neurons, co-immunostainings for vGLUT vs. vGAT or Homer vs. Gephyrin in these iPS cell-derived neurons also revealed very little co-localizations between two different synapse types, again suggesting that glutamatergic vs. GABAergic specifications occupy mutually exclusive pre- and postsynaptic zones (Fig. 6d, and Supplementary Fig. S8f, g). Exogenous GABA-puffs demonstrated sizable inward IPSCs at $V_{hold} \approx -70$ mV that reversed around $\approx 0$ mV, again confirming the presence of surface GABA$_A$Rs also in iPS cell-derived neurons without or with V57 co-expression (Fig. 6e). In continuous recordings from same neurons in the control condition, treatments with CNQX along with NMDA receptor (NMDAR) blocker 3-carboxypiperazin-propyl phosphonic acid (CPP) silenced the majority of synaptic currents regardless of the $V_{hold}$, corroborating their pure glutamatergic identity (Fig. 6f, g). In V57 cells, successive application of PTX and CNQX + CPP respectively inhibited sIPSCs and sEPSCs, attesting for the co-existence of both glutamatergic and GABAergic synapses (Fig. 6f, g). Moreover, holding the V57 neurons at $-70$ mV, but not at $+10$ mV (i.e. near estimated Cl$^-$ reversal-potential, $E_{Cl} \approx +14$ mV, in whole-cell configuration without accounting for junction-potential), depicted authentic mIPSCs even in presence of TTX, as well as triggered reliable evoked IPSCs with robust delayed release and pronounced short-term plasticity (Fig. 6h, i). Hence, our approach was highly reproducible in generating fully operational human GABAergic synapses in vitro, irrespective of reprogrammed stem cell lines.

**Formation of GABAergic synapse in vivo.** We next aimed to investigate if the V57 factors can produce functional GABAergic synapses also in live animals. For this proof-of-principle experiment, we intended to use a model glutamatergic synapse based on two selection criteria: (i) the viral-injection site (i.e., cell bodies of presynaptic neurons) is physically distant from the recording site (i.e., cell bodies of postsynaptic neurons), and (ii) the postsynaptic neuron receives negligible GABAergic inputs from other sources. These constraints were necessary to ensure that viral transduction of V57 factors mainly targets the desired glutamatergic presynaptic neuron, without indirectly potentiating any local GABAergic inputs to the postsynaptic cell.

To this end, we injected virus particles into the mouse spiral ganglion, which gives rise to auditory nerve (AN) fibers that project purely glutamatergic outputs (also known as the 'endbulb of Held') primarily onto the cell bodies of bushy cells (BCs), located distantly in the cochlear nucleus (Fig. 7a)[36,37]. We performed lentiviral injections of V57 factors on postnatal day 2–4 (P2–4), prepared brain-slices and analyzed them on P18–32, i.e. when a control virus encoding RFP showed substantial transduction at the injection site (Supplementary Fig. S9a, b). The BCs lack any well-defined GABAergic input, and mainly receive either Calretinin-positive glutamatergic endbulbs from around 4–5 AN fibers or vGAT-positive glycinergic terminals from local interneurons, that form physically separate synapses from each other (Fig. 7b)[38–41]. However, animals transduced with V57 factors demonstrated prominent increase in co-localization between Calretinin and vGAT signals, indicating a successful induction of transgenes within AN fiber terminals (Fig. 7b, c). Of note, the in vivo transduction efficiency of lentivirus was relatively sparse compared to our in vitro approach (see Fig. 3c, d), and did not significantly alter the endbulb size (Fig. 7b, c).

To determine if ectopic expressions of V57 in otherwise glutamatergic endbulbs can trigger the formation of functional GABAergic synapses, we conducted voltage-clamp recordings from postsynaptic BCs in the presence of Strychnine that preferentially inhibits Glycine receptors (GlyRs) over GABA$_A$Rs. In control condition, the BCs exhibited mostly sEPSCs with fast $\tau$-decays, whereas V57 induction caused a profound rise in sIPSCs with slower $\tau$-decays (Fig. 7d, e). Lentivirus-mediated mosaic transduction of V57 factors did not change either the amplitude or the frequency of sEPSCs, but substantially elevated sIPSC frequency without affecting sIPSC amplitude (Fig. 7f, g).

Finally, we inspected the PSCs of BCs evoked by presynaptic AN stimulation. In control animals, Strychnine-insensitive evoked PSCs showed a rapid decay kinetics, and were largely blocked by acute treatment of AMPAR antagonist NBQX (Fig. 7h, i). However, animals transduced by V57 factors often displayed the co-presence of a slower IPSC component that could only be inhibited by GABA$_A$R blocker Bicuculline (Fig. 7h, i). Several

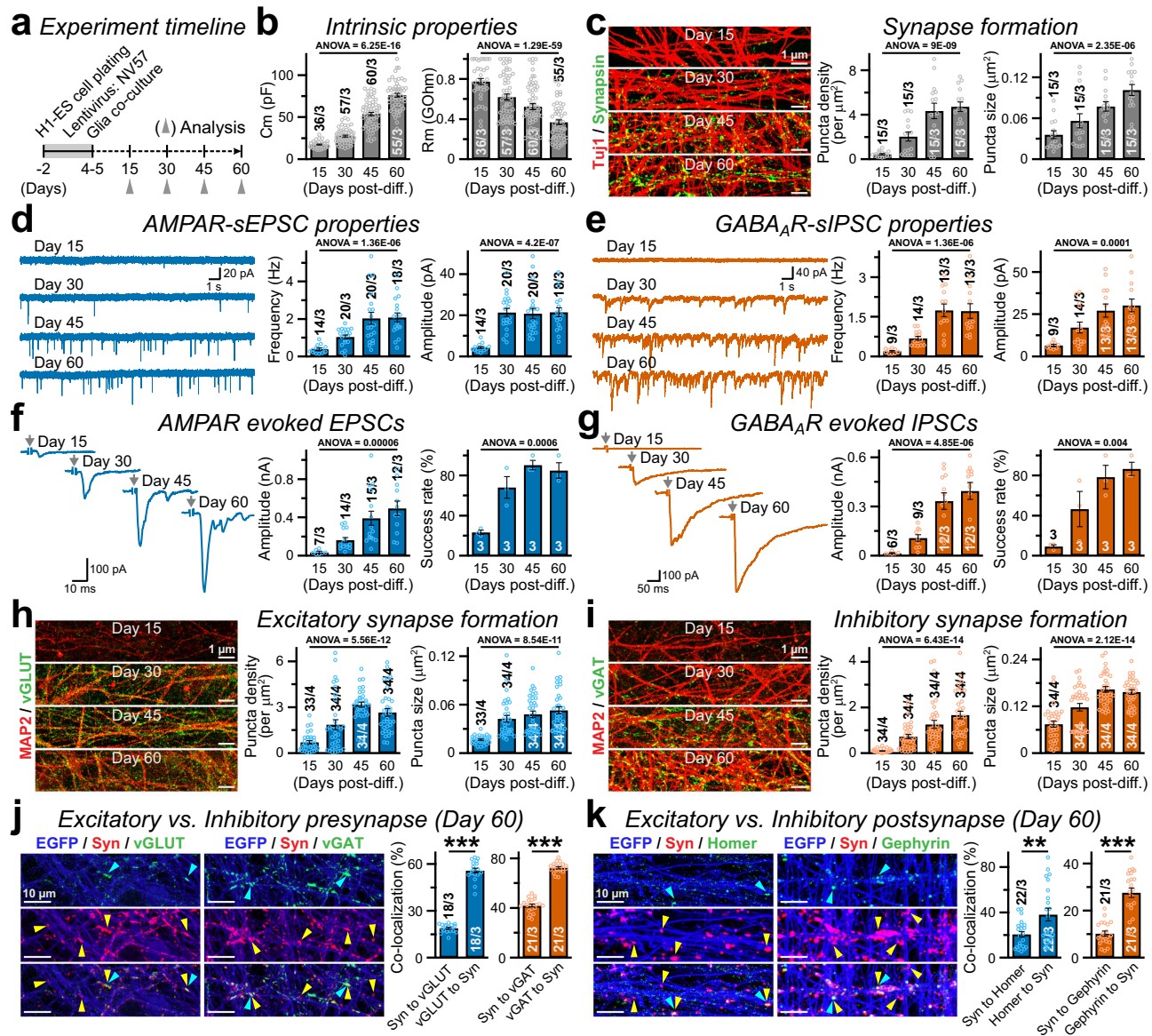

**Fig. 4 Induced GABAergic synapses mature rapidly, in parallel with glutamatergic synapses. a** Experimental protocol for **b–k**; Ngn2-induced human neurons were additionally infected with viruses expressing V57 factors, and analyzed every 15 days starting from post-induction day 15 until day 60. **b** Summary graphs of $C_m$ (left) and $R_m$ (right) values, at different time-points of neuronal maturation. **c** Example images (left) and average density or size (right) of Synapsin puncta constituted on Tuj1-positive neurites, as measured from NV57 neurons at different stages of their in vitro development (see annotated). **d, e** Sample traces (left) and average parameters (right, event frequency or amplitude) of AMPAR-mediated sEPSCs **d** and GABA$_A$R sIPSCs **e** recorded in the presence of PTX and CNQX, respectively, at day 15–60. **f, g** Stimulation-evoked EPSCs (**f**, with PTX) and IPSCs (**g**, with CNQX) recorded at indicated time-points; example traces (left), average amplitudes (middle), and percentage of cells with detectable responses (right). **h, i** Same as **c**, except for vGLUT (**h**) or vGAT (**i**) puncta formed on MAP2-positive dendritic branches. **j**. Representative images (left) and Mander's coefficients (right) of co-localization between Synapsin puncta and either vGLUT or vGAT signals. Neurites were visualized by co-expression of soluble EGFP. Both vGLUT and vGAT signals individually occupy only a fraction of Synapsin-positive total synapses at day 60. **k** Same as **j**, except for co-localization between Synapsin and either Homer or Gephyrin, at day 60. All data represent means ± SEM. Summary graphs also denote the total number of field-of-views analyzed (for immunostaining) or neurons patched (for electrophysiology)/independent batches, and individual data-points (open circles). Statistical significance for normally distributed data (Skewness and Kurtosis values between −2 and 2) in panels **j** and **k** was calculated by two-tailed, unpaired, Student's t-test, with ***$P < 0.005$; **$P < 0.01$. For all group-wise comparisons (time-course, **b–i**), one-way ANOVA was performed, and $P$-values were stated.

BCs in the V57 condition also manifested predominantly evoked IPSCs without any detectable EPSCs, a phenomenon that was never observed in the control condition (Fig. 7i). When averaged from BCs only with any measurable response, V57-induction significantly elevated the amplitude of evoked IPSCs without altering EPSCs (Fig. 7j). Thus, V57 factors can successfully trigger the formation of functional GABAergic synapses also in an in vivo system.

## Discussion

The mammalian central nervous system contains different types of synapses that release and sense a variety of neurotransmitters. The ability to synthesize and transmit these distinct chemicals often requires mutually exclusive sets of enzymes as well as vesicular transporters that are endogenously expressed in certain neuronal lineages. After neuronal fates are established during early development, the type of transmitter produced by a

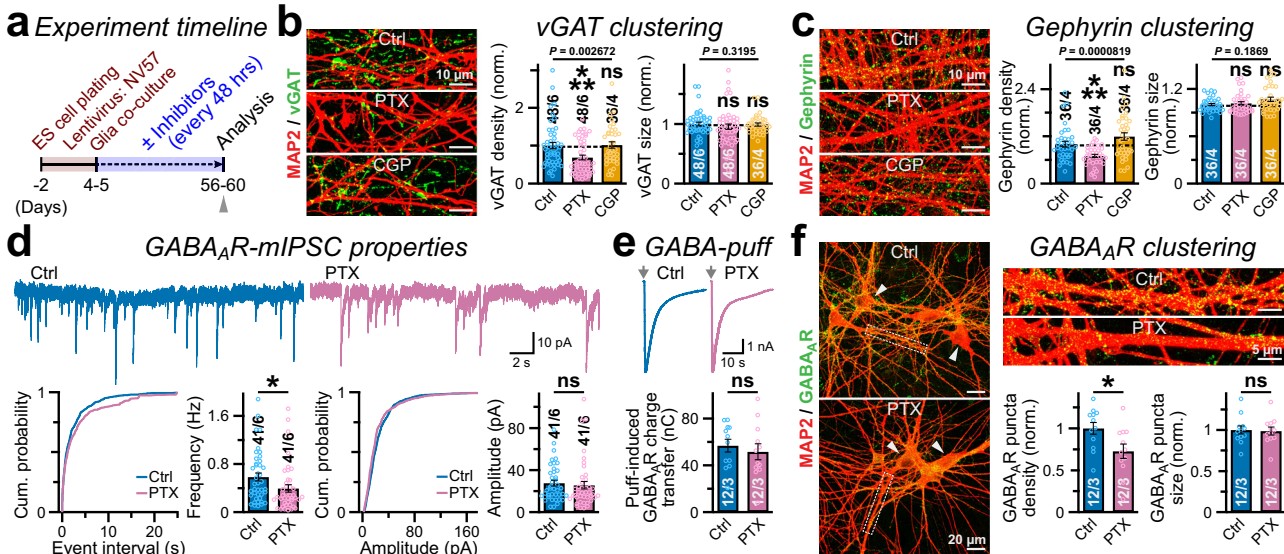

**Fig. 5 GABAergic synapse properties are modulated by GABA$_A$R activity. a** Experimental strategy for panels **b–f**; NV57 neurons were incubated with synaptic inhibitors, half-exchanged media every other day from post-induction day 4–5 to day 56–60, and analyzed afterwards as indicated (arrow). **b, c** Sample images (left) and normalized density or size (right) of vGAT **b** and Gephyrin **c** clusters formed on MAP2-positive dendrites, when treated with DMSO-only (control), 100 μM PTX, or 10 μM CGP55845. **d** Representative mIPSC waveforms (top) and event frequency or amplitude (bottom) plotted as cumulative distribution (left) with summary graphs (right), for control vs. PTX-treated (long-term) neurons. Cultures were washed thoroughly with bath-solution before electrophysiological recordings; CNQX was used to stop EPSCs. **e** Example traces (top) of GABA$_A$R currents produced by 1 mM GABA-puff, and total charge-transfer (bottom). **f** Sample images (left) of control vs. PTX-treated neurons (arrowheads), as immunostained for extracellular epitopes of surface GABA$_A$Rs and dendritic MAP2. Boxed regions are magnified (cropped insets, top right), normalized density and sizes of GABA$_A$R clusters are plotted (bottom right), for control vs. PTX-treatment. All summary data are means ± SEM, with total number of cells recorded (electrophysiology) or field-of-views analyzed (imaging) / independent batches, and individual data-points were included as open circles. A near-normal distribution was predicted for most datasets, based on Skewness or Kurtosis values (-2 >≈ and ≈ < 2). Hence, statistical significance was primarily assessed by two-tailed, unpaired, Student's t-test, with ***$P < 0.005$; *$P < 0.05$; ns = not significant, $P > 0.05$. Multiple groups (**b** and **c**) were compared by Kruskal-Wallis test paired with post-hoc nonparametric Mann-Whitney U-test, and corresponding $P$-values were reported.

presynaptic neuron is considered to be stable and irreversible. Since the transmitter identity is an inherent feature of specific neural subtypes, all synaptic connections formed between a given pair of neurons are generally homotypic, e.g., either glutamatergic or GABAergic, that does not usually change over time[42,43].

We here illustrated that ectopic expression of vGAT + GAD65 + GAD67 can assign robust GABAergic identity on exclusively glutamatergic human and mouse neurons. This presynaptic release of GABA was able to activate postsynaptic GABA$_A$Rs, and produced fully functional GABAergic synapses that manifested bona fide miniature and AP-dependent IPSCs with pronounced short-term plasticity (Figs. 1 and 2). These GABAergic phenotypes were highly reproducible in multiple cell lines, for both in vitro and in vivo systems (Figs. 6 and 7). Direct reprogramming of transmitter type did not alter total synapse number, but caused rearrangement in pre/post-synaptic structures (Fig. 3). The induced GABAergic synapses formed independently of neighboring glutamatergic synapses, but exhibited similar maturation kinetics, and were partially dependent on GABA$_A$R activity (Figs. 4 and 5). In sum, our approach instituted a facile avenue to induce functional synapses by de novo transmitter synthesis (Supplementary Fig. S10a).

Our findings are entirely consistent with a previous report indicating that the ability to release glutamate vs. GABA by different neuronal subtypes could be primarily dependent on their differential expression of transmitter-specific enzymes and vesicular transporters[44]. Once produced, the vesicles containing different transmitters may not necessarily require any specialized release machineries unique to different synapse types. In alignment with this theory, earlier studies have also described co-transmission of different chemicals from the same neurons, and sometimes

even from the same presynaptic terminals with spatially segregated but morphologically similar release sites[21,23,45–49].

Interestingly, although the V57-expressing neurons synthesized both glutamate and GABA, they produced mutually exclusive mEPSC and mIPSC events with distinct kinetics. These in turn could be inhibited individually by selective receptor blockers without attenuating the frequency of other event types (Figs. 1, 2, 6, and 7). These results imply that glutamate and GABA are co-transmitted but likely not co-released simultaneously to co-activate their corresponding receptors. In addition, both pre- and postsynaptic markers labeling glutamatergic vs. GABAergic synapses also manifested limited co-localization (Figs. 3 and 6, and Supplementary Figs. S5, S8f, g), indicating they are physically segregated (Supplementary Fig. S10b). Again, a similar phenomenon was seen in vivo for different co-transmitters, that are often found to be distributed into separate synaptic vesicles and released independently[45,47,50].

How does presynaptic GABA release facilitate the creation of functional synapses? Recent findings suggest that activation of GABA$_A$Rs can play a critical role in this pathway, since genetic deletion of GABA$_A$R subunits or pharmacological inhibition with specific antagonists has been shown to impair GABAergic synapse formation[20,27]. Although in agreement with this notion, long-term PTX exposure also reduced the density of V57-induced GABA-ergic synapses, but it failed to eliminate them entirely (Fig. 5). Furthermore, although V57 factors elevated the density of GABAergic postsynapses, substantial level of both Gephyrin and GABA$_A$R clusters were already present even in Ngn2-only neurons, which lack any presynaptic GABA release. Hence, additional cellular mechanisms might also contribute to the development of GABAergic synapses. For instance, GABA$_A$Rs

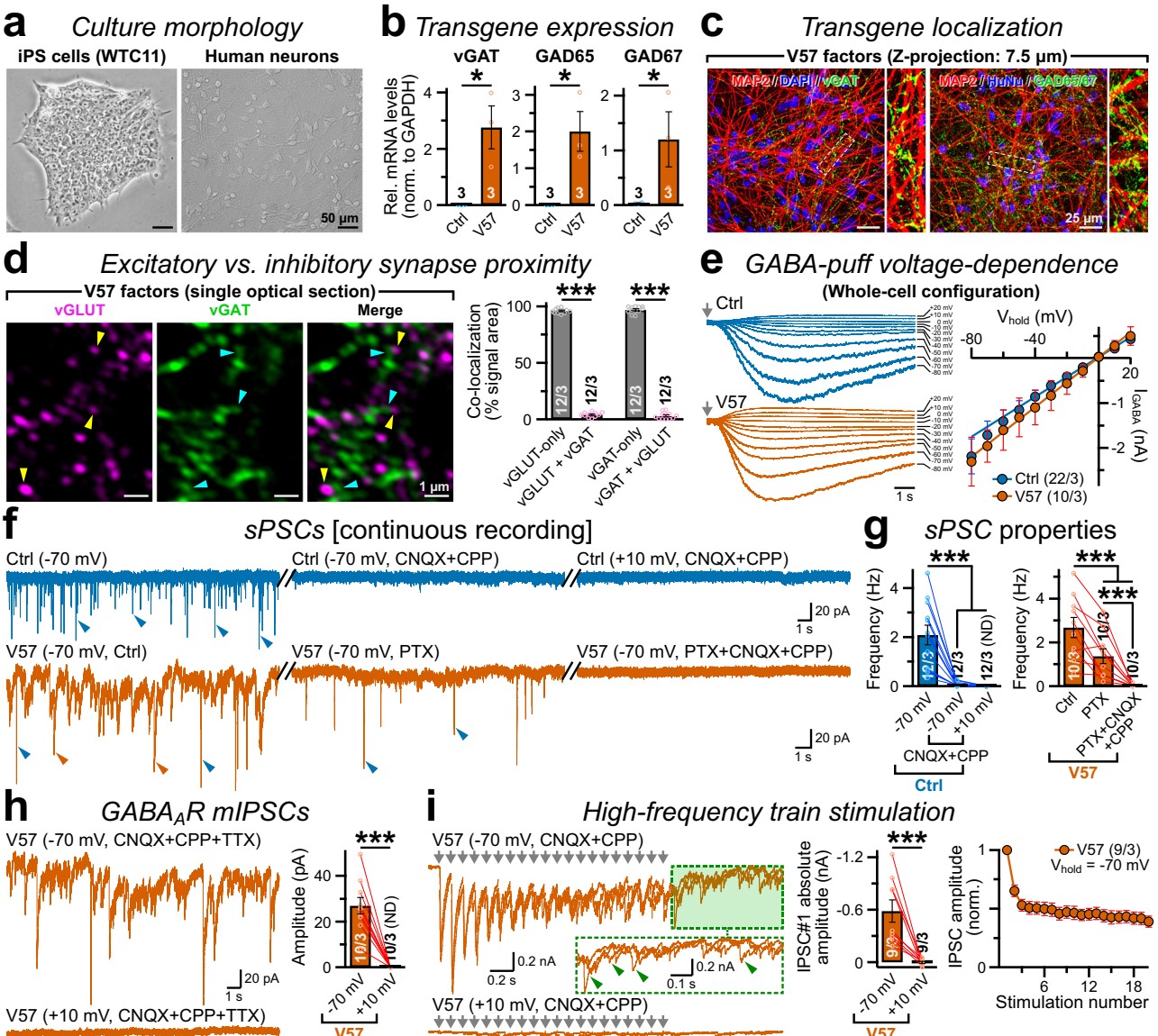

**Fig. 6 V57 factors induce GABAergic synaptogenesis irrespective of cell lines. a** Phase-contrast images of iPS cells (WTC-11 line, left) and differentiated human neurons (day 42, right). **b**, **c** Quantitative RT-PCR (qRT-PCR, **b**) and immunostainings **c** against vGAT or GAD65/67 transgenes in iPS cell -derived neurons, co-labeled for dendritic MAP2, nuclear DAPI, and a human nuclear antigen HuNu. **d** Sample super-resolution image from a single confocal plane (left) and Mander's coefficient values (% area, right) confirm minimal co-localization between vGLUT vs. vGAT puncta (yellow vs. cyan arrows, respectively). **e** Example traces (left) and current-voltage relationships (I-V curves; peak amplitudes at $V_{hold} = -80$ to $+20$ mV, with 10 mV stepwise increments, right) of IPSCs triggered by 1 mM GABA puffs (arrows) on iPS cell-derived neurons, in control and V57 conditions. Average data are fit using straight lines (equation: $y = a + bx$); $a = 32.5 \pm 10$ pA, $b = 22.33 \pm 1.22$ pA for control neurons, and $a = 29.64 \pm 18$ pA, $b = 27.89 \pm 2.7$ pA for V57 condition. **f**, **g** Representative traces **f** and event frequencies **g** of sPSCs recorded at $V_{hold} = -70$ mV or $+10$ mV, in the presence of PTX or CNQX + CPP, in control (top) vs. V57 (bottom) condition, as indicated. Split marks between traces concatenate continuous recordings from same neurons; sEPSCs (blue arrows) or sIPSCs (red arrows). **h** Example traces (left) and average amplitudes of TTX-insensitive mIPSCs (right) recorded at $V_{hold} = -70$ mV or $+10$ mV, in the presence of CNQX + CPP, from iPS cell-derived neurons transduced with V57 factors. **i**. Evoked IPSCs generated by repetitive high-frequency stimulation in V57 condition; overlayed traces (left) with reliable synchronous release and prominent delayed release component (inset, arrowheads with dotted box magnified) after the stimulation train (arrow series) was terminated; average IPSC amplitudes and short-term depression at different $V_{hold}$, as annotated (right). Average values are provided as means ± SEM, with total number of cells recorded (for electrophysiology) or field-of-views analyzed (imaging)/independent batches, or only the number of batches (**b**). All experimentally coupled data-points are provided as color-matched open circles with connected lines. Data distributions were approximately normal (Source Data), for Skewness and Kurtosis values $-2 > \approx$ and $\approx < 2$. Statistical significance was assessed by two-tailed, paired, Student's t-test, with ***$P < 0.005$; *$P < 0.05$; ND = events not detected.

directly interact with presynaptic Neurexins, and can also potentially assemble into molecular complexes with postsynaptic Neuroligins, Gephyrin, and/or Collybistin to establish pre- and postsynaptic alignment with or without GABA release[34,51,52]. Alternatively, GABA molecules may bind to yet unknown trans-synaptic elements to enable inhibitory synaptogenesis. Of note, similar to GABAergic synapses, activation of Glycine receptors (GlyRs) can also promote their synaptic clustering in spinal neurons, which could be mediated via common cell-signaling pathways downstream to receptor activation[53].

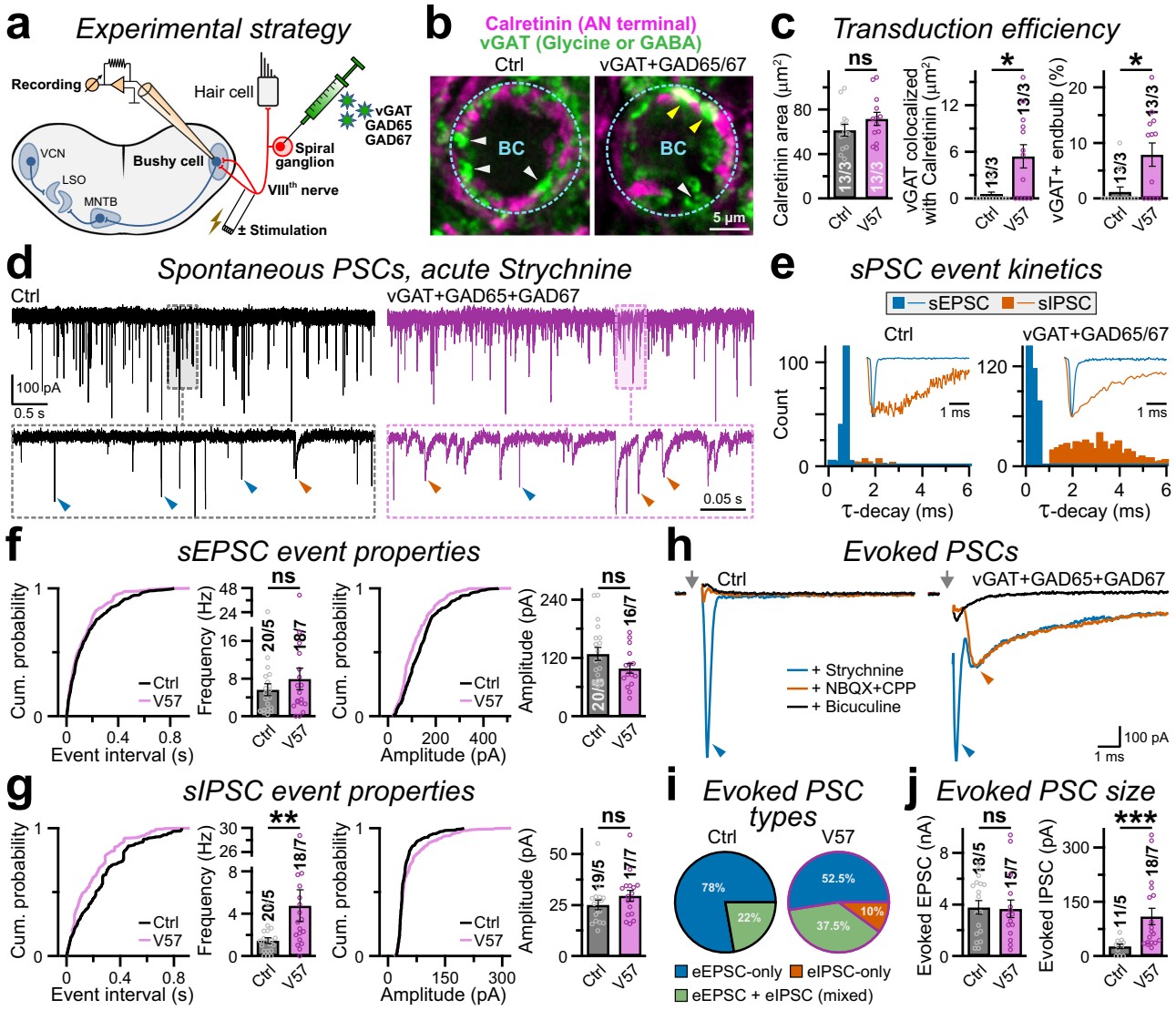

**Fig. 7 V57 factors generate functional GABAergic synapses in vivo. a** Schematic diagram depicts experimental strategy to manipulate the transmitter identity of endbulb synapse. **b** Representative images of BC somas (cyan circles) surrounded by Calretinin and vGAT -positive presynaptic terminals, in control (left) vs. V57-injected (right) animals. The vGAT signals often overlapped with Calretinin-positive endbulb terminals in V57 condition (yellow arrows), but minimally for control condition (white arrows). **c** Average size of AN-terminals measured using Calretinin signals (left), endbulb area co-occupied by vGAT signal (middle), and fraction of endbulbs co-expressing vGAT (right), presumably induced by the V57 factors. **d, e** Sample traces **d** and frequency distribution of τ-decays **e** of sEPSCs (blue arrowheads and histogram) and sIPSCs (red arrowheads and histogram) recorded from control (left) vs. V57 (right) conditions. Insets in **e**, sample sEPSC and sIPSC traces scaled and superimposed to compare their event kinetics. All recordings were performed in the presence of 1 μM Strychnine to suppress sIPSCs from local glycinergic inputs. **f, g** Cumulative probability plots and average bar-graphs of event frequency (left) and amplitudes (right), for AMPAR-mediated sEPSCs **f** or GABA$_A$R-mediated sIPSCs **g**. **h** Example traces of evoked PSCs (stimulus artifacts replaced by grey arrows), recorded from control (left) vs. V57 (right) condition, after acute treatment of 1 μM Strychnine (blue traces), or following the additions of 10 μM NBQX + 5 μM CPP (red traces), and 20 μM Bicuculline (black traces). Arrowheads, EPSC (blue) or IPSC (red). **i** Pie-charts represent fraction of cells with indicated PSC types (either EPSC-only, or IPSC-only, or both). **j** Amplitudes of evoked EPSCs (left) and IPSCs (right), averaged from neurons with successful responses. Averages indicate mean ± SEM, with total number of field-of-views analyzed (for imaging) or neurons patched (for patch-clamp recording) / number of injected animals, and individual data-points (color-coded open circles). Statistical significance was tested by two-tailed, unpaired, Student's t-test (near normally distributed results), or two-sided, nonparametric Mann-Whitney U-test, with ***$P < 0.005$; **$P < 0.01$; *$P < 0.05$; ns = not significant, $P > 0.05$.

Previously, disruptions of presynaptic release machineries were shown to have only minor to no effects on synaptogenesis, as well as dendritic spine formation and their maintenance[54–57]. It is worth recognizing that, although these elegant genetic approaches eliminated the vast majority of synaptic activities, it remains plausible that a minimal level of basal transmitter signal, e.g. residual spontaneous release, is sufficient to institute synapse identity. Even in the absence of a neuronal release, diffused

transmitter secretion from local astroglia cells may also trigger synaptic receptor activation[58]. Alternatively, presynaptic proteins associated with enzymatic production and/or vesicular packaging of different neurotransmitters themselves might directly or indirectly interact with other synaptic molecules and participate in synaptogenic processes, even in the absence of active transmitter release. The contributions of synaptic receptor activation in glutamatergic synapse formation also remain controversial.

Although persistent activation of ionotropic glutamate receptors was shown to facilitate spine outgrowth[59–61], removal of all AMPAR and NMDAR subunits failed to affect presynaptic vesicle distribution or postsynapse morphology[62]. Therefore, multiple parallel pathways could modulate glutamatergic synapse development, and these mechanisms might differ significantly from GABAergic synaptogenesis. Future studies are needed to investigate these various possibilities.

Using glutamate and GABA uncaging, earlier studies have reported that acute neurotransmitter release itself can facilitate nascent synapse formation, in a spatially-delimited fashion[26,27]. Here we demonstrated that, under the physiologically relevant context of presynaptic vesicular release, such transmitter-induced synapses can achieve much greater morphological and functional maturation. Compared to photo-stimulation methods, the V57-induced GABAergic synaptic structures attained substantial growth in their density and size, and displayed reliable IPSCs with high reproducibility, making it amenable for stable, efficient, and extensive modification of synapse identity. Our protocol was also successful in both in vitro model of human neuronal cultures and mouse brain in vivo, and hence, could potentially be adopted to manipulate circuits affected by addiction or mental disorders.

Using small molecule-mediated cellular differentiation and transcription factor screening, we and others have previously derived GABAergic neurons directly from ES cells[63–67]. These differentiated or reprogrammed neurons express various markers and exhibit AP properties that mimic GABAergic neuron subtypes indigenous to the human brain. Although our current approach does not generate specific neural lineages, GABAergic synapses produced by V57 factors similarly manifest robust miniature and evoked IPSCs with comparable amplitude and frequency, that mature over time and can be utilized for in vitro assays. Viral transduction of GABAergic enzymes also offers certain advantages over transcription factor-induced GABAergic neurons, especially for their in vivo applications. When transplanted into the brain, reprogrammed neurons sometime struggle with long-term survival, lack of functional maturation, restricted migration, and limited synaptic integration into pre-existing neural circuits[63–65]. Viral delivery of V57 factors may bypass some of these technical issues by forming new GABAergic synapses in already available neuronal populations.

## Methods

**Institutional approvals**. All cell culture methods as well as lentivirus production procedures were approved by the Institutional Biosafety Committee (IRB protocol # 19-059B) at Colorado State University. All experiments in mice (including male and female, JAX strain CBA/CaJ) were approved by the Institutional Animal Care and Use Committee (IACUC protocol # 201800101) of University at Buffalo, and conducted according to the ethical guidelines.

**Cell lines**. Human ES cells (H1-line, catalog # WA01) were purchased from the WiCell Research Institute under a material transfer agreement (MTA # 19-W0439). The human iPS cell line (WTC-11) was generously gifted by Dr. Michael E. Ward, National Institute of Neurological Disorders and Stroke (NINDS). The human embryonic kidney (HEK) 293T cells for virus production were commercially available from Takara Bio USA (catalog # 632180).

**Viral constructs**. The lentiviral constructs used for neuronal reprogramming of human ES cells included Ngn2-t2A-Puromycin[Resistance] (Tet-on promoter) and rtTA (Ubiquitin promoter), with an optional virus encoding EGFP (Tet-on promoter) for morphological analyses[31]. The cDNAs encoding human vGAT, GAD65, and GAD67 (V57 factors) were cloned into lentiviral vector driven by human Synapsin-1 promoter (see Supplementary Fig. S8b), followed by a Woodchuck Regulatory Element (WRE), and flanked by 5' and 3' long terminal repeats (LTRs).

**Lentivirus production**. Three helper plasmids (i.e., pRSV-REV, pMDLg/pRRE, and VSV-G; 7–8 µg each) and corresponding expression vectors (15–20 µg) were co-transfected with polyethylenimine (PEI) into 70–80% confluent HEK 293T (containing SV40 T-antigen to facilitate virus production) cells plated on 10 cm dishes. At 8–10 h post-transfection, the culture medium was exchanged completely, and the supernatant containing lentivirus particles was collected after 36 h and 60 h. The supernatant was then pooled and spun at ~800 × $g$ for 6–8 min to remove any HEK cell debris. The supernatant was then spun at ~120,000 × $g$ for 2 h at 4 °C (Beckman L8–70M ultracentrifuge equipped with SW41Ti rotor). The viral pellets were resuspended in ~50-100 µl DMEM media, stored overnight at 4 °C, subsequently aliquoted and frozen at -80 °C prior to experimental use.

**Generation of human neurons**. Both the human H1-ES cells (see Figs. 1–5) and an isogenic iPS cell line with doxycycline-inducible Ngn2 transgene (WTC-11; see Fig. 6)[33] were maintained in mTeSR™1 / mTeSR™ Plus media (StemCell Technologies) under feeder-free condition. Media was changed every day. When cell density reached ~70%, they were dissociated with phosphate-buffered saline (PBS) + 0.5 mM EDTA and plated at 1:6 dilution onto Matrigel (BD Bioscience) -coated wells. During the passage, cultures were additionally supplemented with ROCK-inhibitor Y-27632 (2.5 µM, MedChem Express) overnight, but excluded for later media changes.

For H1-ES cells, Ngn2-mediated direct neuronal conversion was achieved as described before[31,32]. In brief, ES cells were co-infected with lentiviruses encoding rtTA and Ngn2-t2A-Puromycin[Resistance], induced with doxycycline (2 µg/ml), selected using puromycin (1 µg/ml), gently dissociated with PBS + EDTA or accutase (Innovative Cell Technologies) and replated with primary mouse glia (passage 1–2, derived from CD-1 ® IGS mice) on Matrigel-coated coverslips (see Fig. 1a). Similar strategy was adopted for iPS cell-derived neurons, that were directly reprogrammed by doxycycline exposure (see Supplementary Fig. S8a)[33]. The neurons were additionally infected with lentiviruses expressing the V57 factors during or immediately after differentiation, as depicted in protocol timelines, and a virus made from empty pFSW-67 vector was used as infection control.

From days 0 to 14, neurons were cultured in N3 media (DMEM/F12 [Thermo Fisher] + N2 [Thermo Fisher] + B27 [Thermo Fisher]), supplemented with insulin [20 µg/ml, Sigma], penicillin/streptomycin [Thermo Fisher]). During glia co-culture, 2–2.5% fetal bovine serum (FBS, Atlas Biologicals) was included. The media was half-exchanged every 3–4 days, and additionally supplemented with 5-fluorodeoxyuridine (FdU, 10 µM) to inhibit glial growth after reaching 70–80% confluency. From day 15 onward, the N3 media was gradually replaced by Neurobasal Plus media [Thermo Fisher] + B27 + penicillin/streptomycin, also supplemented with FBS + FdU.

**In vitro electrophysiology**. Whole-cell patch-clamp recordings of human neurons were performed similarly to that described before[67,68]. In brief, neurons were patched using internal solution containing (for voltage-clamp, in mM) 135 CsCl$_2$, 1 EGTA, 1 NaGTP, 2 QX-314, and 10 HEPES-CsOH (pH 7.4, 310 mOsm); or (for current-clamp, in mM) 130 KCl, 10 NaCl, 2 MgCl$_2$, 0.5 EGTA, 0.16 CaCl$_2$, 4 Na$_2$ATP, 0.4 NaGTP, 14 Tris-creatine phosphate, and 10 HEPES-KOH (pH 7.3, 310 mOsm). The extracellular bath-solution contained (in mM) 140 NaCl, 5 KCl, 3 CaCl$_2$, 1 MgCl$_2$, 10 glucose, and 10 HEPES-NaOH (pH 7.4, 300 mOsm). All recordings were performed at room temperature, using an integrated patch-clamp amplifier (IPA, Sutter Instruments) controlled by customized Igor Pro 8 (Wave-Metrics) data acquisition system. For all cells, the patch quality was monitored using series-resistance values (R$_s$ < 15 MOhm), which did not alter significantly between experimental groups. Voltage-clamp recordings for AMPAR EPSCs and GABA$_A$R IPSCs were conducted at a V$_{hold}$ of −70 mV, unless mentioned otherwise (Fig. 6). Evoked PSCs were triggered by field stimulations using a matrix electrode (FHC, MX21AEW-RT2) connected to an A365RC isolated pulse stimulator (World Precision Instruments). AMPAR- or GABA$_A$R-mediated PSCs were isolated using PTX (100 µM; GABA$_A$R/GlycineR blocker, Tocris Bioscience) or CNQX (25 µM; AMPAR blocker; Tocris Bioscience), respectively. Although all voltage-clamp recordings at V$_{hold}$ = −70 mV contained extracellular Mg$^{2+}$ to block NMDARs, some experiments at V$_{hold}$ = +10 mV (see Fig. 6) also included CPP (50 µM; NMDAR inhibitor; Tocris Bioscience). Tetrodotoxin (TTX; 2 µM; Ascent Scientific) was added to the external solution during all miniature mEPSC and mIPSC recordings, to avoid presynaptic release caused by spontaneous APs. Pressure perfusion of 1 mM AMPA (RS-AMPA hydrobromide, Tocris Bioscience) or 1 mM GABA (Tocris Bioscience) was performed for 100 ms, with 20 psi puffs using Picospritzer III (Parker Instrumentation), and total charge-transfers were calculated within 30 s from puff application.

**Long-term drug incubations**. For Fig. 5b–f, cells were incubated with either GABA$_A$R antagonist PTX (100 µM), or GABA$_B$R antagonist CGP55845 (10 µM, Tocris Bioscience), or DMSO (control) immediately after co-plating with glia, at post-induction day 4–5. Media was half-exchanged every other day with equivalent doses of drugs, and cells were analyzed at day 56–60 (Fig. 5a). For immunostaining, cultures were washed 3–4 times with PBS, fixed by 4% paraformaldehyde (PFA) at room temperature, and processed for antibody incubation (see below). For whole-cell patching, neurons were washed thoroughly 3 ×2 min with bath-solution before recording.

**In vivo mouse injections**. The method of in vivo lentiviral injection into mouse round window was adopted from previous protocols using minor modifications[69,70].

In brief, small incisions were made ~1 cm caudal to the pinna of neonatal mice (P2-P4), and the tympanic bulla was exposed to reveal the round window underneath (see Supplementary Fig. S9a). The tympanic bulla was punctured with a 34-gauge needle, and allowed to drain for at least 10 min. Approximately 0.3 µl of lentivirus (a cocktail of 0.1 µl GAD65, 0.1 µl GAD67, and 0.1 µl vGAT) or AAV-chrimson/RFP (control) was slowly injected into the round window using a 5 µl Hamilton syringe with a 34-gauge removable needle, 0.375" 12° bevel. Note that a higher injection volume could result in liquid overspill, disruption of the cochlear aqueduct, and leakage into cerebrospinal fluid. After retracting the needle, the surgical area was sutured, and the pups were returned to their home cages. Adult animals were sacrificed after 2–4 weeks of viral infection, and brain-slices were prepared for imaging or electrophysiology experiments. Endbulb synapses develop rapidly after birth, attain morphological and functional maturation by ~3 weeks[71–74].

**Brain-slice recordings**. The mice were anesthetized using 200 mg/kg ketamine plus 10 mg/kg xylazine, then sacrificed, brains were removed, and placed into ice-cold sucrose solution (in mM: 76 NaCl, 75 sucrose, 25 NaHCO$_3$, 25 glucose, 2.5 KCl, 1.25 NaH$_2$PO$_4$, 7 MgCl$_2$, 0.5 CaCl$_2$). Sagittal sections (142 µm) were cut using a vibratome (Leica VT1200), and then incubated in standard recording solution (in mM: 125 NaCl, 26 NaHCO$_3$, 20 glucose, 2.5 KCl, 1.25 NaH$_2$PO$_4$, 1.5 MgCl$_2$, 1.5 CaCl$_2$, 4 Na L-lactate, 2 Na-pyruvate, 0.4 Na L-ascorbate, bubbled with 95% O$_2$ - 5% CO$_2$) at 34 °C for 20 min. Afterwards, slices were kept at room temperature until recordings. During recording, 1 µM Strychnine was added to inhibit spontaneous glycinergic IPSCs, 10 µM NBQX and 5 µM CPP were added to respectively block AMPAR and NMDAR -mediated EPSCs, and 20 µM Bicuculine was added to block GABAergic IPSCs. Whole-cell voltage-clamp recordings were made from BCs in AVCN slices using borosilicate patch pipettes of resistance 1.3–2.3 MΩ. Pipettes were filled with internal solution containing (in mM): 35 CsF, 100 CsCl, 10 EGTA, 10 HEPES, and 1 QX-314, pH 7.3, 300 mOsm. BCs were patched under an Olympus BX51WI microscope with a Multiclamp 700B (Molecular Devices) controlled by an ITC-18 interface (Instrutech), driven by custom-written software (mafPC) running in Igor (WaveMetrics). The bath was perfused at 3–4 ml/min using a pump (403U/VM2; Watson-Marlow), with saline running through an inline heater to maintain the temperature at 34 °C (SH-27B with TC-324B controller; Warner Instruments). BCs were held at -70 mV with access resistance 5 to 15 MΩ compensated to 70%. Single presynaptic endbulb terminals were stimulated using a glass micro-electrode placed 30 to 50 µm away from the BC soma with 4–20 µA currents through a stimulus isolator (WPI, A360). Presynaptic stimulation was applied every 8 s. For V57 condition, all sEPSC and sIPSC results were analyzed from neurons only with detectable evoked IPSCs.

**Immunostaining**. Human neuronal cultures with or without V57 factors were fixed in 4% PFA for 30 min at room temperature. Cells were then blocked in 5–10% cosmic calf serum (CCS) for 1 h at 37 °C, incubated with primary antibodies (see Supplementary Fig. S3) for 1–2 h at 37 °C while rocking, washed 4 times with blocking buffer, followed by 1 h incubation at 37 °C with Alexa Fluor (Invitrogen) 488/555/647-conjugated secondary antibodies [488 goat anti-mouse (A11029), 546 goat anti-mouse (A11030), 647 goat anti-mouse (A32728), 488 goat anti-rabbit (A11034), 546 goat anti-rabbit (A11035), 647 donkey anti-rabbit (A31573), 488 goat anti-chicken (A11039), 546 goat anti-chicken (A11040), 647 goat anti-chicken (A21449), 488 goat anti-guinea pig (A11073), 555 goat anti-guinea pig (A21435), or 647 goat anti-guinea pig (A21450)] at 1:1000–2000 dilutions. Cultures were then washed 4 times with blocking buffer and PBS, and the coverslips were mounted upside down on glass slides using Fluoromount-G (Southern Biotech). The cell nuclei were stained with DAPI (1:50000; Thermo Fisher, catalog # D1306) for 5–10 min, when applicable. Most immunostaining assays were performed in a permeabilized environment where Triton X-100 (0.1%) was applied to blocking buffer and for all subsequent steps, including washes or antibody dilutions. However, to visualize cell-surface localization of GABA$_A$Rs (see Fig. 5f, and Supplementary Fig. S6d) and their distributions at dendritic branches, we used a primary antibody against the extracellular epitope of GABA$_A$R subunit α3 under non-permeabilized conditions (without Triton X-100), subsequently permeabilized using Triton X-100, and immunolabeled for dendritic MAP2.

For immunostainings of the cochlear nucleus, mice were transcardially perfused with 0.9% saline followed by 4% PFA, brains were then post-fixed in 4% PFA for an hour and placed in 20% sucrose overnight. Staining of the cochlea involved additional steps, i.e. removing it from temporal bone, decalcification by incubating with 120 mM EDTA for ~5 days, followed by embedding in 100 bloom gelatin, and fixing overnight in PFA. Frozen embedded tissues were cut in 50 µm sections using a microtome, washed 3 times in 0.2 M PBS (0.9% NaCl), blocked with 5% goat serum in PBS + Triton X-100 for 1 h at room temperature, and incubated overnight at 4 °C with primary antibodies (Supplementary Fig. S3). Slices were washed 3 times in PBS and incubated in a solution containing Alexa Fluor (Invitrogen) 568 donkey anti-goat (A11057), 594 goat anti-rabbit (A11037), 488 goat anti-mouse (A11029), and/or 488 donkey anti-rabbit (A21206) secondary antibodies (1:250). Slices were then washed 3 times with PBS and mounted in ProLong diamond antifade mountant (Invitrogen, P36961).

**Image acquisition and analysis**. Confocal images of cultured human neurons were acquired using an inverted STELLARIS 5 (Leica Microsystems) laser scanning microscope and processed with a Leica Application Suite version X (LAS-X, Core_3.7.4_23463) software. Series of optical z-projections were obtained with ~0.5–1 µm optical thickness using either a 20× dry objective or oil-immersion objectives (40× or 60×). All super-resolution images (see Figs. 3f, j, 6d, and Supplementary Fig. S5a, b) were collected (dimension in x/y/z axis: 0.04 × 0.04 × 0.18 µm) using a Zeiss LSM 880 microscope (Zen 2.3 black edition, software v.14.0.9.201) equipped with plan-apochromat 63× oil-immersion objective (1.4 na) and Airyscan Gallium Arsenide Phosphide (GaAsP) detector, that reported to have spatial resolution of 120 nm in x/y- and 350 nm in z-plane[75]. Images of mouse brain tissues, i.e. cochlear nucleus and spiral ganglion neurons (see Fig. 7b and Supplementary Fig. S9b), were captured using an Olympus FV1000 confocal microscope, with ~1.84 µm optical z-sections.

All confocal images were analyzed using FIJI-ImageJ (NIH) software. To quantify various parameters of synaptic puncta along the neuronal processes, images were generally superimposed as maximum-intensity z-projection (10–20 optical slices). Synaptic signals from regions-of-interest (ROIs) were normalized with respect to corresponding neurites areas (MAP2 or EGFP -labeled). Co-localization between two synaptic markers was assessed by first thresholding individual channels appropriately to eliminate background signals for individual experimental batches, and then measuring Mander's coefficients for each optical section within JACoP plugin. For super-resolution images, processing and analysis modules within the ZEN 2.3 (blue edition) software (v.2.3.69.1000) were used to extract maximum-intensity profiles, and measure fluorescence intensities.

**Quantitative RT-PCR**. iPS cell-derived human neurons from the control vs. V57 condition were washed with PBS and collected in 500 µl TRIzol reagent. Immediately, 250 µl chloroform was added to the cell lysate, vortexed vigorously, centrifuged at 12,000 × g for 15 min, aqueous phase collected, and RNA precipitated by adding 250 µl of isopropanol and centrifuging at 12,000 × g for 10 min. The RNA pellets were then washed with 70% ethanol, air dried, and dissolved in nanopure water. cDNA was generated from 300 to 800 ng of total RNA using the Invitrogen SuperScript III First-Strand Synthesis SuperMix (catalog # 11752–050, Thermo Fisher Scientific) following manufacturer's protocol. Quantitative PCR (qPCR) was performed on a CFX-96 (Bio-Rad) machine using SYBR Green Master Mix (catalog # RK21203, ABclonal). All primer sets were designed to span between two adjacent exons, and human GAPDH was used as an internal control (see Supplementary Fig. S8c, d).

**RNA-sequencing dataset**. RNA-sequencing results of Ngn2-induced human neurons (Supplementary Fig. S1c) were previously deposited by us in the NIH database (GEO repository, accession # GSE129241 [https://www-ncbi-nlm-nih-gov.ezproxy.u-pec.fr/geo/query/acc.cgi?acc=GSE129241])[32], and are publicly available.

**Immunoblotting**. Day 56–60 NV57 neurons were collected by scraping, and lysed in RIPA buffer (150 mM NaCl, 5 mM EDTA, 25 mM Tris pH 7.4, 1% Nonidet P-40 substitute, 0.5% Sodium Deoxycholate) supplemented with Halt™ protease inhibitor cocktail (PIC, Thermo Scientific, catalog # 78429). Lysates were mixed at 3:1 with 4x sodium dodecyl sulphate (SDS) loading buffer, run on 7.5% poly-acrylamide gel (PAGE), and then transferred to a nitrocellulose membrane. Membranes were blocked with 3% bovine serum albumin (BSA) in Tris-buffered saline (TBS + 1% Tween-20) for 2–3 h at ambient temperature, and immunostained overnight at 4 °C with primary antibodies. Membranes were subsequently stained with fluorescent secondary antibodies (1:2000 in TBS + Tween-20) for 2–3 h at 37 °C, imaged using LI-COR Odyssey CLx system, and analyzed with Image Studio Lite software (version 5.2).

**Statistics & reproducibility**. For all experimental results, the average values were presented as X/Y, where 'X' represents the total number of neurons recorded (for electrophysiology) or field-of-views analyzed (for imaging) from 'Y' number of independent batches (for human neurons) or animals (mouse brain-slices). All average data indicate means ± SEMs (standard-deviation [SD] of a parameter divided by square-root of number of samples). All samples were chosen randomly, and no data were excluded from analysis. At least >= 3 biological replicates were used, and sample sizes were selected so that SEM ≈ < 1/10$^{th}$ of their respective means for most datasets. Except Figs. 3 and 5, investigators were not blinded to allocation during experiments or outcome assessments, because many assays required prior knowledge of drug identity during acute applications (Figs. 1, 2, 6, and 7), transgene combinations (Figs. 1 and 7), or sample collections and processing at specific time-intervals (Fig. 4).

The numerical values from all figure panels (both main and supplementary) are provided as a Source Data file. For near-normally distributed datasets (i.e., with Skewness and Kurtosis values $-2 > \approx$ and $\approx < 2$), statistical evaluations between conditions were conducted using unpaired (paired for batchwise comparisons), two-tailed, Student's t-test (***$P < 0.005$; **$P < 0.01$; *$P < 0.05$; ns = not significant, $P > 0.05$); otherwise, two-sided, nonparametric Mann–Whitney U-test

was performed as mentioned in the corresponding figure legends. For all groupwise assessments, the *P*-values of single-factor ANOVA (for near-normal data distribution) or Kruskal–Wallis test (for considerable deviations from normal distribution) were reported.

**Reporting summary**. Further information on research design is available in the Nature Research Reporting Summary linked to this article.

## Data availability

Source Data are provided with this paper (see excel file). These include all individual datapoints and average values presented in both the main manuscript (Figs. 1–7) and supplementary information (Figs. S1–S10). The raw data for imaging and electrophysiology experiments are available from the corresponding authors, upon reasonable requests. RNA-sequencing dataset of Ngn2 neurons can be obtained from the publicly available GEO repository (accession # GSE129241), as deposited by our earlier study[32]. Source data are provided with this paper.

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

## Acknowledgements

This work was supported by a start-up fund from the Colorado State University to S.C., and grants from the National Institutes of Health (R01-MH126017 to S.C.; R37-MH052804 to T.C.S.; and R01-DC015508 to M.A.X.). We thank Drs. Robert E. Cohen and Tingting Yao, Colorado State University, for providing us with a Zeiss LSM 880 microscope with Airyscan detector to acquire the super-resolution images of synapses. We also thank Dr. Michael E. Ward, NINDS, for supplying us with the WTC-11 iPS cell line.

## Author contributions

T.C.S. and S.C. conceived the project; S.C. supervised the research; M.A.X., T.C.S., and S.C. designed the experiments; S.R.B., N.F.W., L.P., L.L., C.D.S.P., O.B., M.G., and S.C. conducted the experiments; S.R.B., N.F.W., L.P., O.B., T.P.C. and S.C. analyzed the data; M.A.X., T.C.S., and S.C. wrote the paper; all authors reviewed the manuscript.

## Competing interests

The authors declare no competing interests, either financial or non-financial, in relation to the work described in this study. All requests for experimental reagents should be addressed to S.C. (soham.chanda@colostate.edu).
