## [Peer review file · Nature Communications]

Induction of Synapse Formation by De Novo Neurotransmitter SynthesisREVIEWER COMMENTS

Reviewer #1 (Remarks to the Author):

Burlingham and colleagues investigate the mechanisms underlying synapse identity. They find that forced expression of two synthesizing enzymes and a vesicular transporter of the inhibitory neurotransmitter GABA in human and mouse glutamatergic neurons is sufficient to induce functional GABAergic synapses. The findings are intriguing; the results are well-described, and the experiments are well-performed and largely convincing. A few points need to be addressed before the manuscript can be published:

Major comments

1. **Statistics.** Independent two-sample Student's t-tests are used for all analyses performed in the study, including those where multiple groups are compared (Fig 1e, f, k + Fig 2i + Fig 5b,c + Fig 3e). This should be corrected. The authors should test whether their data is normally distributed for pair-wise comparisons and use the appropriate test, and should perform ANOVA with post-hoc testing for experiments comparing multiple groups.
2. **Fig 3e:** the merged image shows larger presynaptic puncta positive for both vGLUT and vGAT. The authors nicely demonstrate that the postsynaptic elements of glutamatergic and GABAergic synapses are spatially segregated, but it is difficult to judge whether this also applies to the presynaptic elements, as drawn in the schematic in Supp Fig 7. It would be informative to see super-resolution or even electron microscopy images to determine how vesicles and active zones are organized at boutons that appear double-positive for vGLUT and vGAT.
3. **Mouse in vivo experiment (Fig 6).** The authors claim sparse transduction of spiral ganglion neurons, but no data is provided to support this claim. This makes the results difficult to interpret: it is impossible to judge whether the calretinin/vGAT-double positive boutons in Fig 6b are in fact derived from transduced auditory nerve fibers; furthermore, the sparseness of transduction is emphasized (Fig 6b,c), but the contribution of GABAergic postsynaptic currents in the electrophysiological recordings appears substantial. These issues need to be addressed.

Minor comments

1. **Fig 6:** please explain the rationale for the use of the spiral ganglion/auditory nerve as model system. Is there a specific reason this experiment could not be attempted in for instance the hippocampal circuit, such as the glutamatergic Schaffer collateral synapse in CA1?
2. **Fig 6a:** the image should be replaced with a clear schematic outlining the details of the synaptic connection being studied.
3. **Fig 6b, c:** please explain in the Results why it is important that transduction is relatively sparse and does not impact endbulb size.
4. The authors cite studies showing that manipulating GABA A receptors impairs morphology and target specificity of GABAergic synapses in both Introduction and Discussion. It would be useful to also discuss studies in which glutamatergic synaptic transmission was silenced (e.g. Lu et al., Neuron 2013/PMID:23664612). Would the experiment the authors did also work for the induction of glutamatergic synapses in GABAergic neurons?
5. **Fig 1:** it would be helpful to discuss how the properties of GABAergic outputs from human ESCs differentiated into inhibitory neurons compare to the ones induced by vGAT+GADs treatment in Ngn2 neurons.
6. The point where the authors indicate that this research could help "treat encephalopathies" seems a bit too far-fetched and is unnecessary in my opinion.

Reviewer #2 (Remarks to the Author):

Burlingham et al. investigated mechanisms of synapse formation by using stem cell derived human neurons. Ectopic expression of GABA synthesis enzymes and vesicular transporters was sufficient to release GABA. This GABA release caused increased inhibitory synaptic responses and inhibitory synaptic proteins. In vivo injection of lentiviruses expressing V57 factors showed similar effects as found in culture neurons. Evidence for increased inhibitory synapse number was analyzed mostly by patch clamping recording and immunostaining. The suggested conclusions sound interesting, but they were based on observations without having a mechanistic understanding. The electrophysiology experiments also need better clarification. Details are listed below.

Main criticisms:

1. To demonstrate purely glutamatergic neurons (by Ngn2), cell was recorded at -70 mV and PTX and CNQX were applied. Because amplitudes of IPSCs and EPSCs are largely affected by a membrane potential and chloride reversal potential, this approach will not surely discard the possibility of GABA release. IPSCs could still be detected in other conditions. It may be better to record at +10 mV (using the same internal solution) and examine if the IPSCs are still not detected in the presence of NBQX (or CNQX) and D-APV.

2. In figure 1, analyzing IPSCs by slower decay kinetics and selective elimination of these slow components by PTX is somewhat confusing. As mentioned above, the size and kinetics of PSCs could be altered by resting membrane potentials, series resistance, and chloride reversal potentials, judging the identity of responses by decay kinetics cannot be conclusive evidence. Representative traces shown in Fig 1g also do not look as if they originated from the same cell. The authors need to show continuous recording traces from the same cell by the following conditions:

In one set of experiments, measure PSCs in the presence of D-APV at -70 mV. After baseline recording, apply PTX and continue to measure PSCs. Series resistance should be kept low and stable without compensation. Uninterrupted continuous traces before and after PTX will demonstrate both a condition of recording and response changes.

If all responses with slower decay kinetics are eliminated by PTX at -70 mV, analyze chloride reversal potentials to see if the big amplitudes of IPSCs detected at -70 mV can be explained by them.

In another set of experiments, cells should be held at +10 mV in the presence of D-APV. After getting a baseline, apply CNQX to see if similar findings are observed. Recording traces should be plotted in a continuous mode before and after CNQX application to check if there are no changes in holding currents and series resistance.

The resting membrane potentials and series resistance of all recorded neurons should also be plotted together with above results.

3. In Figure 2f-h, mPSC recording also needs to be performed as described in comment #2. Distinguish EPSCs and IPSCs separately. Analyzing only decay kinetics will not provide decisive evidence for their identities.

4. Figure 3f-h showed that Homer and Gephyrin are not colocalized. Just calculating the percentage of 'Homer only' and 'Gephyrin only' does not give rigorous quantitative information. Plot a distribution curve showing the overall distance between Homer and Gephyrin puncta. Does it fit well with presynaptic vGLUT and vGAT distribution (Fig 3b-e)?

5. sIPSCs shown in Figure 4e and 4g should be measured at +10 mV in the presence of NBQX or CNQX.

6. Figure 5 showed that GABA receptor activation is critical for the formation of GABAergic

synapses. Is this because GABA is depolarizing? Because figures 1 and 2 showed huge inward currents that seem to be mediated by GABA_A receptor activation, examining this possibility will be important. Can GABA puff trigger postsynaptic Ca²⁺ rise?

7. IPSCs seem to be repetitively measured at -70 mV holding potentials (Figure 6d). p18-32 neurons are regarded as mature neurons, but very strong IPSCs can be detected at -70 mV. Was this happening only in stem cell derived neurons?

8. As previously mentioned, evoked PSCs (Figure 6e, h) should be measured at +10 mV.

Minor comments

1. In figure 4b, show representative traces.

2. In page 2, "reception" should be "receptor".

3. In page 3, "... of AP-independent authentic mPSCs" should be "mPSPs".

Reviewer #3 (Remarks to the Author):

This study by Burlingham, Wong et al seeks to address a fundamental question in neuroscience—is the expression of a particular neurotransmitter sufficient for creation of an appropriate type of synapse? The question has considerable implication for theories of synaptogenesis, which range from emphasis on the primacy of synaptic adhesion molecule interactions to ideas and data that support causality of neurotransmitter release. Most of experiments supporting the latter theory used exogenous neurotransmitter photolysis to demonstrate causality. Here, the authors take one step back further and impose neurotransmitter release capabilities de novo on a cell type free of components needed for GABA release. They found that ectopic expression of GABA synthesis enzymes and vesicular transporters is sufficient for GABA production and release, as well as GABAergic synapse formation. The work clearly defines the (remarkably small) minimal set of presynaptic proteins sufficient to drive downstream synaptogenesis in response to specific neurotransmitter release. This study is simply and beautifully designed and executed, and the conceptually impactful paper is written with exceptional clarity rarely seen in my experience reviewing for this and other journals. I have a few minor comments.

- How was the spiral ganglion and auditory nerve connection to bushy cells chosen as the test bed for in vivo experiments? There are a number of other cell classes with selective neurotransmitter expression that could have been targeted. (Note, however, that I am not requesting experiments generalizing to other cell types and would consider such requests beyond scope).

- Compared to the quality of the writing, the figures could use refinement. They are very overcrowded, not colorblind friendly, and the wide range of font sizes will ensure that the smallest are hard or impossible to see.

- Considering the importance of uncaging evoked synaptogenesis experiments to the framework of the study, it would be appropriate to expand citations that capture a broader age range and cell types where these phenomena have been demonstrated (example papers not cited: PMC4551606, DOI: 10.1016/j.biopsych.2020.12.022, PMC4716836)

- Finally, the authors should expand consideration of literature demonstrating formation of functional dendritic spines and synapses in the absence of presynaptic neurotransmitter release (e.g., PMC5418202, which is not cited).

- There are insufficient details on statistics and it looks like in some cases the stats may be inappropriate. Was normality assessed? For non-normal or too small data sets non-parametric analog tests are needed. Further, students t-tests seem to be used throughout even when data appear appropriate for one way or two way ANOVA (e.g., Fig 2 I, Fig 3 E, H, among others). Clarify and correct throughout.

Authors' response to reviewers' comments for “**Induction of Synapse Formation by De Novo Neurotransmitter Synthesis**” [NCOMMS-21-38007-T; Burlingham, Wong, and Peterkin et al.], and changes made in the revised manuscript

We thank the reviewers for their careful evaluation of our paper. We truly appreciate the thoughtful and constructive suggestions, which helped to further improve this study. In order to address the reviewers' comments, we have performed a number of additional experiments and amended the text as described in detail below. These new experiments, analysis of existing data, and textual changes have broadened the scope and strengthened the impact of this study. Below we quote the *reviewers' comments in black font with italic typeface*, and provide our point-by-point responses in blue font with regular typeface. We hope that with these additions and changes, this revised manuscript will be acceptable for publication.

Reviewer #1 (Remarks to the Author):

Burlingham and colleagues investigate the mechanisms underlying synapse identity. They find that forced expression of two synthesizing enzymes and a vesicular transporter of the inhibitory neurotransmitter GABA in human and mouse glutamatergic neurons is sufficient to induce functional GABAergic synapses. The findings are intriguing; the results are well-described, and the experiments are well-performed and largely convincing. A few points need to be addressed before the manuscript can be published:

We appreciate this reviewer's positive comments on the conceptual advances and technical merits of our study. We have now incorporated his/her recommendations on statistical analyses, co-localization assays between synaptic markers, and provided rationale for our *in vivo* model, as discussed below.

Major comments

1. Statistics. Independent two-sample Student's t-tests are used for all analyses performed in the study, including those where multiple groups are compared (Fig 1e, f, k + Fig 2i + Fig 5b,c + Fig 3e). This should be corrected. The authors should test whether their data is normally distributed for pairwise comparisons and use the appropriate test, and should perform ANOVA with post-hoc testing for experiments comparing multiple groups.

Agree. We have now performed (i) Student's t-test for near-normally distributed data, (ii) Mann-Whitney U-test for nonparametric statistics, as well as (iii) single-factor ANOVA with post-hoc Tukey-Kramer test and (iv) Kruskal-Wallis test paired with post-hoc U-test and Bonferroni correction, to compare between multiple groups, as described in all figure legends. Please note that, although these new analyses made the statistical interpretation of our results much stronger, they did not change the general conclusions.

2. Fig 3e: the merged image shows larger presynaptic puncta positive for both vGLUT and vGAT. The authors nicely demonstrate that the postsynaptic elements of glutamatergic and GABAergic synapses are spatially segregated, but it is difficult to judge whether this also applies to the presynaptic elements, as drawn in the schematic in Supp Fig 7. It would be informative to see super-resolution or even electron microscopy images to determine how vesicles and active zones are organized at boutons that appear double-positive for vGLUT and vGAT.

This is a great point. We have now used super resolution microscopy (Zeiss Airyscan detector, reported optical resolution: 120 nm in x/y, 350 nm in z-plane) to gain better structural insight into the few synaptic puncta that appeared double-positive for both vGLUT and vGAT in V57-expressing neurons, especially for maximum intensity z-projection of thick optical sections. We found that almost all puncta that showed co-localization between glutamatergic and GABAergic pre- or postsynapses on the x/y-plane, could be further resolved into multiple physically segregated boutons on z-dimension, which no longer supported

the co-existence of these two distinct synapse types (Figs. 3f, 3j, 6d, and Supplementary Figs. S5, S8). These data corroborate that glutamatergic and GABAergic synapses do not co-form at same locations.

3. *Mouse in vivo experiment (Fig 6). The authors claim sparse transduction of spiral ganglion neurons, but no data is provided to support this claim. This makes the results difficult to interpret: it is impossible to judge whether the calretinin/vGAT-double positive boutons in Fig 6b are in fact derived from transduced auditory nerve fibers; furthermore, the sparseness of transduction is emphasized (Fig 6b,c), but the contribution of GABAergic postsynaptic currents in the electrophysiological recordings appears substantial. These issues need to be addressed.*

Since vGAT, GAD65, and GAD67 proteins preferentially localize at presynaptic terminals, we could not directly measure their transduction efficiency in the cell bodies of spiral ganglion neurons. Instead, we now provide representative images from our *in vivo* experiment with RFP-expressing control virus, that demonstrate substantial but mosaic transduction at the injection site (Supplementary Fig. S9b).

Because each factors of the V57 pool individually contributed to $\approx 1/3^{\text{rd}}$ of the total virus volume, we anticipate even less viral delivery and infection per GABAergic genes. Indeed, we found that only $\approx 5\text{-}10\%$ of the endbulb terminals, although significantly higher than control condition, were co-labeled with vGAT (Fig. 7c). Based on these observations, we concurred that our approach produced partial infection of the target cells *in vivo*. We are now in the process of improving this efficiency for a follow-up study.

Minor comments

1. *Fig 6: please explain the rationale for the use of the spiral ganglion/auditory nerve as model system. Is there a specific reason this experiment could not be attempted in for instance the hippocampal circuit, such as the glutamatergic Schaffer collateral synapse in CA1?*

We now discuss the rationale for selecting the endbulb synapse for our current study, in Results section: For this proof-of-principle experiment, we intended to use a model glutamatergic synapse based on two selection criteria: (i) the viral-injection site (i.e., cell bodies of presynaptic neurons) is physically distant from recording site (i.e., cell bodies of postsynaptic neurons), and (ii) the postsynaptic neuron receives negligible GABAergic inputs from other sources. These constraints were necessary to ensure that V57 transduction mainly targets the desired glutamatergic presynaptic neuron, without potentiating any local GABAergic inputs to the postsynaptic cell.... A schematic diagram (Fig. 7a) also illustrates these points.

The auditory nerve (AN) fibers originate from spiral ganglia and project purely glutamatergic outputs (also known as the 'endbulb of Held') primarily onto the cell bodies of bushy cells (BCs), that are located distantly in the cochlear nucleus. The BCs also lack well-defined GABAergic inputs, and mostly receive glycinergic terminals from local interneurons. This unique anatomical structure of the endbulb synapse provided an optimal environment to probe our central hypothesis in *in vivo* system. Since our approach was successful (this study), we are indeed currently adopting similar strategies to generate GABAergic synapses in various other brain regions (including hippocampal circuits, subject of a future publication), with minor modifications in construct design and virus types, that target specific neural lineages.

2. *Fig 6a: the image should be replaced with a clear schematic outlining the details of the synaptic connection being studied.*

We now include a schematic of the synapse studied, illustrating injection and recording sites (Fig. 7a).

3. *Fig 6b, c: please explain in the Results why it is important that transduction is relatively sparse and does not impact endbulb size.*

The low infection efficiency was not intended, but resulted from technical difficulties. Although we used the *endbulb of Held* as a model synapse (see above for rationale, Minor Comment #1) to test our results *in vivo*, this system did not allow us to inject more than 0.1 μl lentivirus per V57 factors. A higher volume (to achieve higher infection efficiency) could result in disruption of cochlear aqueduct and leakage into cerebrospinal fluid causing non-specific spread of virus. We now mention this in the Methods section.

4. The authors cite studies showing that manipulating GABA A receptors impairs morphology and target specificity of GABAergic synapses in both Introduction and Discussion. It would be useful to also discuss studies in which glutamatergic synaptic transmission was silenced (e.g. Lu et al., Neuron 2013/PMID:23664612). Would the experiment the authors did also work for the induction of glutamatergic synapses in GABAergic neurons?

Agree. We now discuss the effects of both activation and silencing of glutamate receptors in excitatory synaptogenesis: “The contributions of synaptic receptor activation in glutamatergic synapse formation also remain controversial. Although persistent activation of ionotropic glutamate receptors was shown to facilitate spine outgrowth, removal of all AMPAR and NMDAR subunits failed to affect presynaptic vesicle distribution or postsynapse morphology. Therefore, multiple parallel pathways might modulate glutamatergic synapse development, and these mechanisms could differ significantly from GABAergic synaptogenesis”. We have also included this important citation, PMID 23664612, amongst others.

We now also cite a relevant work for glycinergic synapses [PMID: 9565032]: “similarly to GABAergic synapses, Glycine receptor (GlyR) activation can also promote its synaptic clustering in spinal neurons, which could be mediated via common signaling pathways downstream to receptor activation.”

Induction of glutamatergic synapse in GABAergic neurons may need additional steps; both knocking out/down vGAT, GAD65, and/or GAD67 expression, as well as overexpression of vGLUT1/2/3. We feel these manipulations will require substantial optimization and are beyond the scope of our current paper.

5. Fig 1: it would be helpful to discuss how the properties of GABAergic outputs from human ESCs differentiated into inhibitory neurons compare to the ones induced by vGAT+GADs treatment in Ngn2 neurons.

Great point. We have now included a section that describes both the advantages and limitations of our current approach over GABAergic neurons differentiated from ESCs (Discussion, paragraph 8): “Using small molecule -mediated cellular differentiation and transcription factor screening, we and others have previously derived GABAergic neurons directly from ES cells. These differentiated or reprogrammed neurons express various markers and exhibit AP properties that mimic GABAergic neuron subtypes indigenous to human brain. Although our current approach does not generate specific neural lineages, GABAergic synapses produced by V57 factors similarly manifest robust miniature and evoked IPSCs with comparable amplitude and frequency, that mature over time and can be utilized for in vitro assays. Viral transduction of GABAergic enzymes also offers certain advantages over transcription factor -induced GABAergic neurons, especially for their in vivo applications. When transplanted in brain, reprogrammed neurons sometime struggle with long-term survival, lack of functional maturation, limited migration, and synaptic integration into existing circuits. Viral delivery of V57 factors may bypass these technical issues by forming new GABAergic synapses in already available neuronal population.”

6. The point where the authors indicate that this research could help “treat encephalopathies” seems a bit too far-fetched and is unnecessary in my opinion.

We have now removed this sentence from our revised manuscript.

Reviewer #2 (Remarks to the Author):

Burlingham et al. investigated mechanisms of synapse formation by using stem cell derived human neurons. Ectopic expression of GABA synthesis enzymes and vesicular transporters was sufficient to release GABA. This GABA release caused increased inhibitory synaptic responses and inhibitory synaptic proteins. In vivo injection of lentiviruses expressing V57 factors showed similar effects as found in culture neurons. Evidence for increased inhibitory synapse number was analyzed mostly by patch clamping recording and immunostaining. The suggested conclusions sound interesting, but they were based on observations without having a mechanistic understanding. The electrophysiology experiments also need better clarification. Details are listed below.

We also thank this reviewer for his/her positive comments on the potential interests of our findings. We now address the technical concerns regarding our electrophysiological recordings. We feel that majority of these comments resulted from lack of experimental details provided in our original manuscript, which we now included, and also conducted a number of additional tests that further justify our interpretations.

Main criticisms

1. *To demonstrate purely glutamatergic neurons (by Ngn2), cell was recorded at -70 mV and PTX and CNQX were applied. Because amplitudes of IPSCs and EPSCs are largely affected by a membrane potential and chloride reversal potential, this approach will not surely discard the possibility of GABA release. IPSCs could still be detected in other conditions. It may be better to record at +10 mV (using the same internal solution) and examine if the IPSCs are still not detected in the presence of NBQX (or CNQX) and D-APV.*

We respectfully disagree. Please note that, although we failed to detect any reliable miniature or evoked IPSC events from Ngn2-only neurons at $V_{\text{hold}} = -70$ mV (Supplementary Fig. S1a), robust IPSCs were readily produced by puff-applications of GABA (Supplementary Fig. S1bii), when recorded at the same V_{hold} , using the same internal-external solution. These results suggest that GABA_ARs are fully functional in Ngn2 neurons under our experimental conditions, but could not be activated directly by a presynaptic GABA release. These data were also supported by the limited expressions of presynaptic proteins that are required for GABA biosynthesis and vesicular release (i.e. vGAT, GAD65/67; see RNA-sequencing results in Supplementary Fig. S1c, and immunostaining results in Fig. 3c, d) in Ngn2-only neurons.

To further convince this reviewer (and ourselves), we have now derived current-voltage relationship (I-V curve, Fig. 6e) for GABA puff -induced IPSCs while holding the neurons (both Ngn2-only and NV57 cells) at different voltages. We found that the absolute IPSC amplitude at $V_{\text{hold}} = -70$ mV is much larger than at +10 mV, with Cl⁻ Nernst potential $\approx +14$ mV, mathematically calculated from ionic concentrations of our internal and external solutions. Moreover, CNQX+CPP inhibited most synaptic currents in Ngn2-only neurons at -70 mV, and holding the same cells at +10 mV did not reveal any underlying GABAergic synaptic activity (Fig. 6f, g). Hence, Ngn2-only neurons are predominantly glutamatergic.

2. *In figure 1, analyzing IPSCs by slower decay kinetics and selective elimination of these slow components by PTX is somewhat confusing. As mentioned above, the size and kinetics of PSCs could be altered by resting membrane potentials, series resistance, and chloride reversal potentials, judging the identity of responses by decay kinetics cannot be conclusive evidence. Representative traces shown in Fig 1g also do not look as if they originated from the same cell. The authors need to show continuous recording traces from the same cell by the following conditions:*

Please note that the event kinetics (half-width and τ -decay) was essential as a screening parameter for initial characterization of two distinct PSC types, without making specific assumption about their identity (Fig. 1c-f). Subsequent application of pharmacological agents in V57-induced neurons demonstrated the presence of both CNQX-sensitive EPSCs with fast τ -decay and PTX-sensitive IPSCs with slow τ -decay, both recorded using same experimental conditions, i.e., same V_{hold} and internal-external solutions (Fig. 1g-k). Upon request, we now also provide continuous PSC traces recorded from the same neurons and applied CNQX, CPP, or PTX (Fig. 6), that further support our previous conclusions (please see below).

In one set of experiments, measure PSCs in the presence of D-APV at -70 mV. After baseline recording, apply PTX and continue to measure PSCs. Series resistance should be kept low and stable without compensation. Uninterrupted continuous traces before and after PTX will demonstrate both a condition of recording and response changes.

If all responses with slower decay kinetics are eliminated by PTX at -70 mV, analyze chloride reversal potentials to see if the big amplitudes of IPSCs detected at -70 mV can be explained by them.

We have now performed similar experiments as suggested (Fig. 6f, g). In uninterrupted recordings from V57-transduced neurons, we first applied PTX and then CNQX+CPP, which successively inhibited the sIPSCs and sEPSCs, again confirming the presence of both GABAergic and glutamatergic synapses.

Cl⁻ reversal potential was also determined in response to Main Criticism #1 by this reviewer (see Fig. 6e). Given the composition of our external and internal solutions (see Methods), $V_{\text{hold}} = -70$ mV (but not +10 mV) produced robust IPSCs (both miniature and evoked) with large amplitude that were convenient to measure (see Fig. 6h, i). We hope that the reviewer will now be satisfied with these observations.

In another set of experiments, cells should be held at +10 mV in the presence of D-APV. After getting a baseline, apply CNQX to see if similar findings are observed. Recording traces should be plotted in a continuous mode before and after CNQX application to check if there are no changes in holding currents and series resistance. The resting membrane potentials and series resistance of all recorded neurons should also be plotted together with above results.

Please see above; $V_{\text{hold}} = +10$ mV is not optimal for IPSC recordings under our experimental conditions.

3. *In Figure 2f-h, mPSC recording also needs to be performed as described in comment #2. Distinguish EPSCs and IPSCs separately. Analyzing only decay kinetics will not provide decisive evidence for their identities.*

Selective inhibitions of glutamatergic vs. GABAergic synapse activities, respectively by CNQX vs. PTX, suggested the existence of both synapse types, whereas co-application of CNQX + PTX eliminated all synaptic currents (Fig. 2f-h). Thus, we also used pharmacological agents, and not only τ -decay.

Upon request, we have now also conducted continuous recordings of TTX-insensitive mIPSCs from V57-induced neurons, that could not be inhibited by CNQX+CPP (Fig. 6h). Again, $V_{\text{hold}} = +10$ mV was not optimal for mIPSC recordings under our experimental conditions for reasons discussed earlier.

4. *Figure 3f-h showed that Homer and Gephyrin are not colocalized. Just calculating the percentage of 'Homer only' and 'Gephyrin only' does not give rigorous quantitative information. Plot a distribution curve showing the overall distance between Homer and Gephyrin puncta. Does it fit well with presynaptic vGLUT and vGAT distribution (Fig 3b-e)?*

Thanks. This is a great suggestion, and certainly a better way of plotting our data. We now present the distributions of glutamatergic vs. GABAergic pre- and postsynapses using line-intensity profiles, which continued to indicate that these two synapses exhibit minimal co-localization (Supplementary Fig. S8f, g). In a few instances, especially for thicker z-stacks with maximum intensity projections, we did notice some overlap between two channels (Fig. 3e, i). However, using super-resolution microscopy (see our reply to Reviewer #1, Major Comment #2), when we mapped signal distributions from individual optical sections, we again found that most co-localized pixels could be further resolved into two or more distinct puncta from different synapse types, in x/z and/or y/z dimensions (Figs. 3f, 3j, Supplementary Fig. S5).

5. *sIPSCs shown in Figure 4e and 4g should be measured at +10 mV in the presence of NBQX or CNQX.*

Please see above; $V_{\text{hold}} = +10$ mV is not optimal for IPSC recordings under our experimental conditions. Also, although we didn't explicitly mention this in our original manuscript, but all IPSC recordings in Fig. 4e, g were indeed performed in the presence of CNQX, whereas EPSC recordings in Fig. 4d, f included PTX. Sorry about this confusion; we've now provided these information in corresponding figure legends.

6. *Figure 5 showed that GABA receptor activation is critical for the formation of GABAergic synapses. Is this because GABA is depolarizing? Because figures 1 and 2 showed huge inward currents that seem to be mediated by GABAA receptor activation, examining this possibility will be important. Can GABA puff trigger postsynaptic Ca²⁺ rise?*

GABA_AR activation does trigger postsynaptic Ca²⁺ rise in Ngn2-neurons (Ca²⁺ imaging data not shown). However, long-term treatments of several Ca²⁺ channel blockers adversely affect cell health and reduce culture longevity. These technical issues prevented us from investigating the role of this Ca²⁺ influx in GABAergic synapse formation, in this paper. We are currently using alternative approaches to address this question, as well as examining the role of synaptic cell-adhesion molecules in this pathway. These experiments will take significant amount of time, and can be subject of a future study. We hope that the reviewer will find our current results (i.e. ectopic expression of vGAT+GAD65+GAD67 in glutamatergic neurons can generate fully functional GABAergic synapses, both *in vitro* and *in vivo*) already interesting.

7. IPSCs seem to be repetitively measured at -70 mV holding potentials (Figure 6d). p18-32 neurons are regarded as mature neurons, but very strong IPSCs can be detected at -70 mV. Was this happening only in stem cell derived neurons?

Please notice that all data in Fig. 7 (previously Fig. 6) were obtained from mouse neurons (the endbulb synapse, *in vivo*) that were virally transduced with V57 factors. Unlike the ES/iPS cell derived and Ngn2-induced human neurons (Figs. 1-6), that require additional time to gain neuronal identity, morphological maturation, and functional properties, the endbulb synapse (presynaptic auditory nerve fiber terminals on postsynaptic bushy cell) in mouse start developing as early postnatal day P0-P1, and already mature by P18-21. Thus, adult P18-32 animals are sufficient for reliable synaptic analyses in this *in vivo* system. We now mention this and cite relevant studies [PMID: 16497457, 12052904, 20107122, 33851098].

8. As previously mentioned, evoked PSCs (Figure 6e, h) should be measured at +10 mV.

Again, V_{hold} = +10 mV is not optimal for our experimental conditions for reasons we discussed above.

Minor comments

1. In figure 4b, show representative traces.

Please note that these (C_m and R_m) values were automatically generated by the oscilloscope window, using a -10 mV test-pulse (20 ms, at 10 Hz) that monitors our whole-cell patch quality. We make a note of these values without necessarily saving the continuous traces from live membrane-test readings.

2. In page 2, “reception” should be “receptor”.

Thanks, we have now revised this sentence.

3. In page 3, “... of AP-independent authentic mPSCs” should be “mPSPs”.

Thanks, we have now corrected this typo.

Reviewer #3 (Remarks to the Author):

This study by Burlingham, Wong et al seeks to address a fundamental question in neuroscience—is the expression of a particular neurotransmitter sufficient for creation of an appropriate type of synapse? The question has considerable implication for theories of synaptogenesis, which range from emphasis on the primacy of synaptic adhesion molecule interactions to ideas and data that support causality of neurotransmitter release. Most of experiments supporting the latter theory used exogenous neurotransmitter photolysis to demonstrate causality. Here, the authors take one step back further and impose neurotransmitter release capabilities de novo on a cell type free of components needed for GABA release. They found that ectopic expression of GABA synthesis enzymes and vesicular transporters is sufficient for GABA production and release, as well as GABAergic synapse formation. The work clearly defines the (remarkably small) minimal set of presynaptic proteins sufficient to drive downstream synaptogenesis in response to specific neurotransmitter release. This study is simply and beautifully designed and executed, and the

conceptually impactful paper is written with exceptional clarity rarely seen in my experience reviewing for this and other journals. I have a few minor comments.

We are grateful to this reviewer for commenting that our study is “beautifully designed” and “written with exceptional clarity”. We also very much appreciate his/her acknowledgement of the conceptual impact of our work. We now address his/her remaining minor points regarding the (i) rationale for using endbulb synapse as *in vivo* model, (ii) statistical analyses, and (iii) included relevant citations (please see below).

- How was the spiral ganglion and auditory nerve connection to bushy cells chosen as the test bed for in vivo experiments? There are a number of other cell classes with selective neurotransmitter expression that could have been targeted. (Note, however, that I am not requesting experiments generalizing to other cell types and would consider such requests beyond scope).

This particular point needed further clarification in our manuscript, and was similarly raised by Reviewer #1. Please see our response to Reviewer #1, Minor Comment #1 (justification added in Result section).

- Compared to the quality of the writing, the figures could use refinement. They are very overcrowded, not colorblind friendly, and the wide range of font sizes will ensure that the smallest are hard or impossible to see.

We used consistent font size for all images. The apparent variability in font size could have arisen from JPEG versions of the images (used in the initial submission) which automatically zoomed in for smaller figures and zoomed out for larger figures. We have now uploaded PDF versions of all figures that reflect original font sizes. Upon request, we have now used colorblind friendly colors for all graphs in all figures.

- Considering the importance of uncaging evoked synaptogenesis experiments to the framework of the study, it would be appropriate to expand citations that capture a broader age range and cell types where these phenomena have been demonstrated (example papers not cited: PMC4551606, DOI: 10.1016/j.biopsych.2020.12.022, PMC4716836).

We have now included all 3 suggested citations (in Introduction) to further highlight the significance of earlier uncaging-induced synaptogenesis experiments, that provided a framework for our current study.

- Finally, the authors should expand consideration of literature demonstrating formation of functional dendritic spines and synapses in the absence of presynaptic neurotransmitter release (e.g., PMC5418202, which is not cited).

This is an important point. We indeed elaborated on this in our Discussion section (see paragraph #6), which also included this highly relevant citation (Reference #56, PMID: 5418202, Sigler et al., 2017).

- There are insufficient details on statistics and it looks like in some cases the stats may be inappropriate. Was normality assessed? For non-normal or too small data sets non-parametric analog tests are needed. Further, students t-tests seem to be used throughout even when data appear appropriate for one way or two way ANOVA (e.g., Fig 2 I, Fig 3 E, H, among others). Clarify and correct throughout.

Agree. We have now used more appropriate statistical tests for all datasets throughout the manuscript, as recommended (also see our response to Reviewer #1, Major Comments #1). However, please note that these new tests did not necessarily change our overall conclusions.

REVIEWERS' COMMENTS

Reviewer #1 (Remarks to the Author):

The authors have convincingly addressed my concerns. Their revisions to the manuscript have further improved their very interesting study, which is ready for publication in my view.

Reviewer #2 (Remarks to the Author):

The revised dataset is much more convincing and electrophysiology recording is clarified.

Figure 6e. "Puff IPSC" should be replaced by "Puff GABA" or something like that.

Reviewer #3 (Remarks to the Author):

The authors have thoroughly addressed my review points and those of other reviewers. I have no further comments.

Authors' response to reviewers' comments for "**Induction of Synapse Formation by De Novo Neurotransmitter Synthesis**" [NCOMMS-21-38007A; Burlingham, Wong, and Peterkin et al.], and changes made in the revised manuscript

We are glad to see the reviewers' general interest and enthusiasm about our revised manuscript. Below we quote *their new comments in black font with italic typeface*, and provide our point-by-point responses in blue font with regular typeface.

Reviewer #1 (Remarks to the Author):

The authors have convincingly addressed my concerns. Their revisions to the manuscript have further improved their very interesting study, which is ready for publication in my view.

We truly thank this reviewer for providing highly constructive suggestions, which further improved the technical strengths of our paper.

Reviewer #2 (Remarks to the Author):

The revised dataset is much more convincing and electrophysiology recording is clarified.

We very much appreciate this reviewer's thoughtful evaluation of our study, which was extremely helpful in revising the manuscript.

Figure 6e. "Puff IPSC" should be replaced by "Puff GABA" or something like that.

We have now changed this figure title according to the reviewer's suggestion.

Reviewer #3 (Remarks to the Author):

The authors have thoroughly addressed my review points and those of other reviewers. I have no further comments.

We are grateful to this reviewer for his/her valuable comments, which significantly broadened the context and scope of our manuscript.